# INTRANSIGENT TEACHERS GUIDE BETTER TEST-TIME ADAPTATION STUDENTS

## ABSTRACT

Test-Time Adaptation has recently emerged as a promising strategy that allows the adaptation of pre-trained models to changing data distributions at deployment time, without access to any labels. To address the error accumulation problem, various approaches have used the teacher-student framework. In this work, we challenge the common strategy of setting the teacher weights to be an exponential moving average of the student by showing that error accumulation still occurs, but only on longer sequences compared to those commonly utilized. We analyze the stability-plasticity trade-off within the teacher-student framework and propose to use an intransigent teacher instead. We show the surprising result that not changing any of the weights of the teacher model within existing test-time adaptation methods allows them to significantly improve their performance on multiple datasets with longer scenarios. Finally, we show that the proposed changes are applicable to different architectures and experimental setups, and are more robust to changes in hyper-parameters.

## 1 INTRODUCTION

Machine learning models typically assume that training and testing data originate from a similar distribution. However, in real-world applications, distribution shifts between training (source) and testing (target) data domains are common and can lead to performance issues throughout inference (Geirhos et al., 2019; Hendrycks & Dietterich, 2019; Koh et al., 2021). Test-Time Adaptation (TTA) (Wang et al., 2021) is an emerging paradigm that allows for an online adaptation of a pre-trained model to the changing data distributions during testing, where there is a lack of access to any labels. While many methods have been developed in recent years (Gong et al., 2022; Goyal et al., 2022; Marsden et al., 2024; Niu et al., 2022; 2023; Rusak et al., 2022; Sun et al., 2020; Wang et al., 2022; Yuan et al., 2023b), important challenges remain within TTA, such as adaptation over very long scenarios (Press et al., 2023), robustness to noisy data (Gong et al., 2023), and sensitivity to hyper-parameter change (Boudiaf et al., 2022; Zhao et al., 2023; Cygert et al., 2024).

The teacher-student paradigm is a popular TTA framework (Chen et al., 2022; Döbler et al., 2022; Sójka et al., 2023; Wang et al., 2022; Yuan et al., 2023b; Zhou et al., 2024), where the teacher weights are set to the exponential moving average (EMA) of the student weights. This strategy, follows pioneering works in semi-supervised learning (Laine & Aila, 2017; Tarvainen & Valpola, 2017), representation learning (Grill et al., 2020; He et al., 2020; Oquab et al., 2024) and learning under label noise (Liu et al., 2020; Nguyen et al., 2020). The averaged model provides more accurate and consistent predictions which the student uses for training. However, this strategy does not necessarily prevent error accumulation, which may result in model collapse (i.e., falling below the accuracy of the source model). In this work, we present experimental evidence indicating that state-of-the-art TTA methods that utilize teacher-student framework (Chen et al., 2022; Wang et al., 2022; Yuan et al., 2023b) are prone to significant accuracy degradation on extended test sequences.

We observe that using a simple technique of making the teacher more intransigent (not updating the teacher's weights) prevents model collapse over very long testing sequences and, interestingly, can teach students to fulfill the age-old cliche of surpassing their teacher (see Fig. 1). Based on that finding, we take a closer look at the stability-plasticity trade-off (Mermillod et al., 2013) within teacher-student frameworks and how it affects the final model performance. We show that while increased teacher plasticity can lead to better performance in the short run, using a more stable teacher

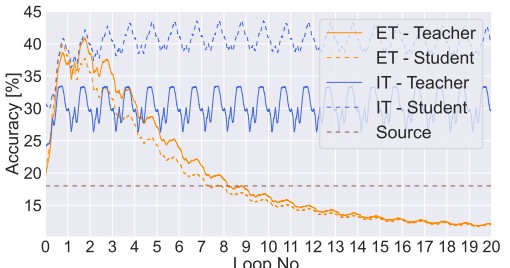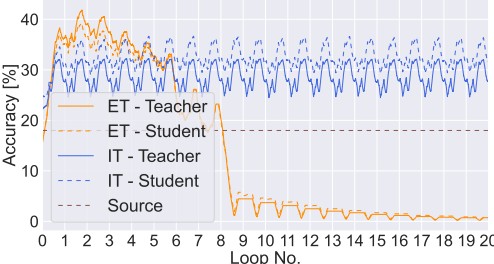

Figure 1: Per-batch accuracy on repeated ImageNet-C (20 loops) using ResNet50 architecture comparing AdaContrast **(left)** and RoTTA **(right)** with well-known EMA teacher (ET, orange) and proposed intransigent teacher (IT, blue), both for teacher (solid) and student (dashed). The EMA teacher visibly suffers from error accumulation as the sequence gets longer. After the 3rd loop, IT manages to avoid model collapse, and surprisingly, the student performs significantly better.

leads to a better adaptation over longer scenarios. This is due to the inevitable error accumulation of the plastic model, which adapts well to the current data but fails to maintain generalization over time, as explored in Sec. 4.2.

We evaluate the use of intransigent teachers over a wide variety of scenarios, incorporating very long adaptation sequences and different experimental settings. The results show consistent improvements over the EMA teacher, significantly reducing the possibility of model collapse. Our contributions are as follows:

- We analyze the EMA teacher framework commonly used in TTA and show its tendency to *not* prevent the model from collapsing on longer test sequences.
- We observe that the Intransigent Teacher (IT) technique, which maintains the teacher's weights fixed during adaptation, effectively prevents model collapse while allowing the student to surpass the teacher.
- The described strategy can be combined with state-of-the-art methods, mitigating error accumulation by easily replacing any EMA teacher and obtaining reliable performance.

## 2 RELATED WORK

**Test-time adaptation**. TTA (Wang et al., 2021) aims to adapt a pre-trained model to shifting data distributions during testing, without any labels. Different unsupervised objectives can be used, examples of which include entropy minimization (Niu et al., 2022; 2023; Wang et al., 2021), cross-entropy variants (Döbler et al., 2022; Wang et al., 2019; 2022) or self-supervised objectives (Sun et al., 2020; Chen et al., 2022). Since the objective is unsupervised, optimizing it over multiple iterations might result in error accumulation (Chen et al., 2019), which is a great challenge in test-time adaptation. Therefore, numerous strategies have been developed to circumvent that issue, amongst of which is the teacher-student framework (Chen et al., 2022; Wang et al., 2022; Yuan et al., 2023b; Sójka et al., 2023). The teacher-student framework was introduced to TTA by the CoTTA method (Wang et al., 2022), where they proposed to use an exponential moving average (EMA) of weights. However, since the teacher is an EMA of student weights, nothing prevents error accumulation in the long run, and therefore we analyze the teacher-student framework in this work.

Other works have explored strategies that keep the fixed source model. ROID (Marsden et al., 2024) continually ensembles weights of a fixed source model with those of a backpropagated model to stabilize the backpropagated model update. Both GROTTA (Li et al., 2023) and TRIBE (Su et al., 2024) incorporate a third, fixed source model for additional regularization with an extra loss term. While these methods employ complex designs, we explore how a simple intransigent teacher technique can address the performance degradation problem in TTA, especially for longer sequences.

Parallel work (Zhou et al., 2024), also proposes to evaluate existing TTA methods over very long adaptation scenarios. Our works are complementary, since they propose an adaptive method to work

in such conditions (which requires parameter tuning), while we focus on evaluating existing approaches on a more extensive experimental setup, and observe that a simpler modification improves the reliability in such conditions.

**Plasticity-stability trade-off.** EMA ensemble of weights is parametrized by a $\beta$ parameter , which determines the balance between retaining old averaged weights of the teacher and incorporating newly updated student ones, commonly set in TTA to 0.999 (Wang et al., 2022) (following temporal ensembling works (Laine & Aila, 2017)), where $\beta = 1$ means full stability (frozen model), and $\beta = 0$ means maximal plasticity. The trade-off has been widely analyzed in continual learning (Mermillod et al., 2013; Chaudhry et al., 2018; Masana et al., 2022), where the learner needs to balance learning of new tasks with the risk of forgetting the previously acquired knowledge. While a variety of strategies have been developed in continual learning, the most successful ones are those that promote stability. Many recent works freeze the feature extractor and learn only the classification part (Goswami et al., 2024; Ma et al., 2023; Panos et al., 2023). Note that TTA can be considered more challenging due to unavailability of labels. Our work is inspired by those studies and aims to analyze the plasticity-stability trade-off within TTA.

**Teacher-student framework.** In knowledge distillation (Hinton et al., 2015), a usually larger model (teacher) guides the optimization of the target model (student) by providing informative training signals (teachers' outputs). Such a strategy is also commonly used in continual learning to prevent forgetting of previous tasks when learning new ones (Buzzega et al., 2020; Kirkpatrick et al., 2017). Self-distillation is a special case in which the teacher and student have the same architecture. It has been shown that in such a scenario, the student can outperform its teacher (Furlanello et al., 2018). In their work, the teacher is updated at the end of every training epoch, by copying students' weights. They show improvement gains until such a procedure is repeated three times. Note that in TTA, there is no access to labeled data, and therefore, updating the teacher might result in even more significant error accumulation.

## 3 INTRANSIGENT TEACHER

The teacher-student framework is widely studied in TTA, with great results being obtained by popular methods such as AdaContrast (Chen et al., 2022) and CoTTA (Wang et al., 2022). Although these methods also rely on other components (e.g., memory queue or weight restoration), they share a common trend of incorporating a self-supervision loss. We dissect their self-supervised losses (directly related to learning), which allows for disentangling the impact on the teacher-student framework by the component that is most related to the stability-plasticity trade-off.

We probe standard EMA teachers (ET) as proposed in their original works (both using $\beta = 0.999$) and compare them with the presented intransigent teacher (IT) technique ($\beta = 1$). Intransigent teachers have all trainable parameters frozen. Their batch normalization statistics are calculated on a per-batch basis (Schneider et al., 2020), as commonly done in TTA (Niu et al., 2022; 2023; Wang et al., 2021; 2022), and the final predictions are taken from the student model. As a motivation example, we evaluate on the popular ImageNet-C corruption benchmark (Hendrycks & Dietterich, 2019) and on the recently introduced CCC benchmark (Press et al., 2023), to observe the adaptation performance over long sequences. Furthermore, we introduce the setting with ImageNet-C repeated 20 times (ImageNet-C (L)), in order to have another very long sequence similar to the classic TTA scenarios. Table 1 shows numerical results and Fig. 2 shows accuracy over time, where the evaluated objectives are described as:

- Consistency: CoTTA (Wang et al., 2022) minimizes the cross-entropy consistency between predictions from the teacher and the student. The input of the teacher is transformed via additional augmentation following the original implementation.

- Contrastive: AdaContrast (Chen et al., 2022) uses a MoCo-inspired (He et al., 2020) contrastive task in which features from different views of the same image (positive pairs) are pulled closer, while features from different images (negative pairs) are pushed away by pseudo-labels. Input from both teacher and student is augmented by randomly drawing two strong augmentations.

Table 1: Mean accuracy [%] of student models on test-time adaptation benchmarks. ET stands for exponential moving average teacher, and IT indicates the intransigent teacher. ImageNet-C (L) stands for the ImageNet-C adaptation sequence being repeated 20 times.

| Loss | Teacher | ImageNet-C | ImageNet-C (L) | CCC |
|------|---------|------------|----------------|-----|
| Source | none | 18.0 | 18.0 | 16.8 |
| Consistency | ET | 27.3 | 7.9 | 1.6 |
|  | IT | 28.8 | 31.3 | 27.2 |
| Contrastive | ET | 35.5 | 22.1 | 5.8 |
|  | IT | 35.4 | 36.9 | 31.8 |

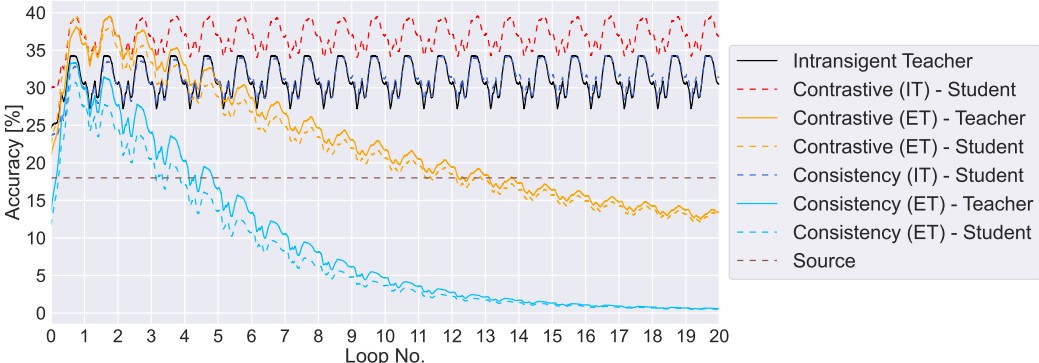

Figure 2: Per-batch accuracy on repeated ImageNet-C (20 loops) with EMA teacher (ET) and intransigent teacher (IT), both for teacher (solid) and student (dashed). Using only the self-supervised losses, without any additional components (or restart mechanisms), allows for successful adaptation over long sequences when the teacher is intransigent.

The results from the above experiment lead to the following observations:

**Observation 1:** Self-supervised objectives combined with intransigent teachers provide great reliability on their own. This is a very positive result, as recent work (Press et al., 2023) observed that on CCC, most of the current adaptation methods result in model collapse. This is especially interesting in the case of AdaContrast, where the contrastive loss changes only the backbone of the model without changing the classification layer, mostly relying on feature alignment.

**Observation 2:** An exponential moving average on its own does not prevent error accumulation. As clearly shown in Fig. 2, the problems when using EMA take some time to become apparent and are only visible over long sequences. In both consistency and contrastive cases, performance degradation starts rather early, after the 2nd and 3rd loops, respectively.

**Observation 3:** When using the EMA, there is a small gap between the teacher and student accuracies. When an intransigent teacher with consistency loss is used, the student performance is reliable and comparable to the teacher, while in the case of using the contrastive loss, the student is able to outperform their teacher significantly.

## 4 METHODOLOGY

As presented in the previous section, using an intransigent teacher with self-supervised objectives provides great model consistency throughout longer scenarios without encountering catastrophic error accumulation. In this section, we aim to better understand the plasticity-stability trade-off within existing TTA teacher-student frameworks, this time when considering all proposed components. First, we describe how to extend existing TTA methods with an intransigent teacher strategy. After, we evaluate and discuss the effects of low-to-none plasticity within the proposed adapted methods.

## 4.1 Appointing an intransigent teacher

We extend popular TTA methods, AdaContrast (Chen et al., 2022), CoTTA (Wang et al., 2022), RoTTA (Yuan et al., 2023a), and PETAL (Brahma & Rai, 2023), with our proposed intransigent teacher. We only modify the value of the $\beta$ parameter used in an EMA ensemble of weights by setting it to 1, and take the final output predictions from the student model, regardless of which prediction was used in the original method. We do not alter the usage or adaptation of batch normalization statistics. Below, we briefly describe the strategies extended with IT.

**AdaContrast** conducts weight updates using a three-part loss function: cross-entropy loss , diversity regularization, and contrastive loss. Pseudo-labels are refined by keeping a buffer of previous image features and their predictions. Refined predictions are based on the nearest neighbors of the current feature within the buffer. Statistics in batch normalization layers are updated with EMA.

**CoTTA** updates the student model by minimizing the cross-entropy consistency between the teacher and the student predictions. Depending on the prediction confidence, the pseudo-labels are the result of averaging predictions on multiple, differently augmented images. At each iteration, there is a small probability for each of the student's weights to be reset to the source pre-trained value. It calculates batch normalization statistics on a per-batch basis.

**RoTTA** keeps the class-balanced memory buffer of images and uses it to perform the optimization in constant intervals. The loss is weighted based on how long the sample has been stored. Cross-entropy-based consistency between the student and teacher models is utilized for a loss function. Batch normalization statistics are updated via EMA.

**PETAL** is similar to CoTTA but it enhances the learning objective by the regularizer term based on a posterior distribution over a source model weights and a data-dependent prior. Moreover, it improves CoTTA's stochastic model reset scheme with the Fisher Information Matrix.

## 4.2 Effects of intransigence

To better understand the plasticity-stability trade-off, we evaluate performance when varying $\beta \in [0.9, 1.0]$, where 1.0 means using an intransigent teacher, and 0.999 is the default value for the two original methods we compare: AdaContrast and CoTTA.

We evaluate on CIFAR10-C and ImageNet-C (Hendrycks & Dietterich, 2019), with each common adaptation sequence repeated 20 times (L). The average final performance is presented in Table 2, and the accuracy evolution through the sequence in Figure 3. Overall, results indicate a clear tendency for intransigent teachers to guide more consistent students, with more lenient teachers performing worse than just using the source model. When allowing for more plasticity (by decreasing the $\beta$ parameter), students improve in the short-run, although inevitably collapsing in the long term. Further, using EMA outperforms the intransigent teacher when some plasticity is allowed ($\beta = 0.9999$). However, some degradation seems to appear towards the end of the sequence. Therefore, a teacher with fixed weights might guarantee a more consistent and reliable performance. Nonetheless, allowing the teacher model to update weights with carefully adjusted EMA parameters might still improve the results further. It should be noted that hyperparameter selection for TTA is problematic (Boudiaf et al., 2022; Zhao et al., 2023). In real-world applications, predicting different

Table 2: Mean accuracy [%] with different exponential moving average $\beta$ parameter for averaging teacher weights. AdaContrast and CoTTA default originally to 0.999. (L) stands for the the adaptation sequence being repeated 20 times. Source does not update the model at all.

| Dataset | Method | 0.9 | 0.99 | 0.999 | 0.9995 | 0.9999 | Intransigence |
|---------|--------|-----|------|-------|--------|--------|---------------|
| CIFAR10-C (L) | Source | - | - | - | - | - | *56.5* |
| | AdaContrast | 79.0 | 79.3 | 81.9 | 83.2 | **85.8** | 85.4 |
| | CoTTA | 10.5 | 14.9 | 55.9 | 68.6 | **78.3** | 68.4 |
| ImageNet-C (L) | Source | - | - | - | - | - | *18.0* |
| | AdaContrast | 1.2 | 5.4 | 18.8 | 25.6 | 38.6 | **40.4** |
| | CoTTA | 0.3 | 23.6 | 52.8 | **55.1** | 50.9 | 35.4 |

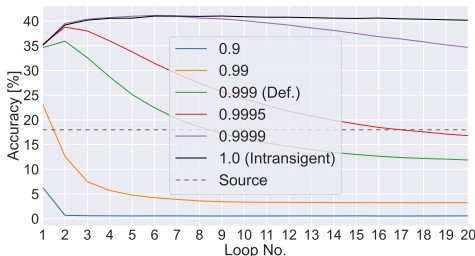 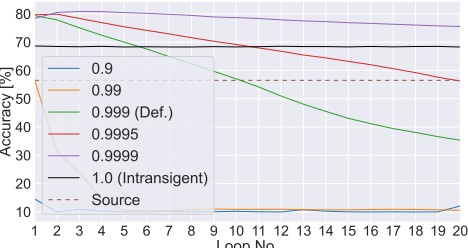

Figure 3: Mean accuracy [%] for each loop of common testing sequence on ImageNet-C (L) with AdaContrast **(left)** and CIFAR10-C (L) with CoTTA **(right)**. The brown dashed line indicates the source model accuracy as a reference.

domain shifts encountered during test time is almost impossible. Figure 3 shows that the performance of different EMA parameters is also highly dependent on the sequence length. Regardless of how promising the initial accuracy is, it can degrade over time.

## 5 EXPERIMENTAL SETUP

**Benchmarks**. We focus on the continual test-time adaptation setting introduced in (Wang et al., 2022), where methods are continually evaluated on a stream of unlabelled test data, without utilizing any knowledge about domain changes. We evaluate the methods on a wide variety of benchmarks with domain shifts. Our experiments include popular corruption benchmarks (Hendrycks & Dietterich, 2019) - CIFAR10-C and ImageNet-C. They involve training the source model on clean CIFAR10 (Krizhevsky, 2009)/ImageNet (Deng et al., 2009) images and testing the adaptation on corrupted images. There are 15 types of corruptions with 5 levels of severity. We follow the setup from (Wang et al., 2022; Niu et al., 2022; Döbler et al., 2022) and use the standard corruption sequence with the highest severity level. The adaptation on natural domain shifts is tested utilizing DomainNet-126 (Peng et al., 2019) and ImageNet-R (Hendrycks et al., 2021) datasets. DomainNet-126 includes data from 4 distinct domains: real, clipart, painting, and sketch. We pre-train the model on a real domain and experiment on the remaining ones. ImageNet-R is composed of 30,000 images portraying different renditions of 200 ImageNet classes.

Our goal is to focus on very long adaptation sequences to evaluate the methods in terms of stability during long-time operation. For that reason, we repeat the standard test sequences of benchmarks described above and loop them 20 times. We call this setup, a Long (L) scenario. Our longest sequence of 3,000,000 images is generated using CIFAR10-C benchmark. Simply repeating the benchmarks 20 times provides limited data variability. Nevertheless, we argue that if existing methods do not cope well in such settings (as we show), they especially would not work well in more complex real-world settings.

Additionally, we take advantage of CCC (Press et al., 2023) benchmark. It was created by applying 2 corruptions from the corruption benchmarks to images from the ImageNet (Russakovsky et al., 2015) dataset. It greatly fits our experiments, since one of the assumptions of this benchmark was to make it very long. We use a CCC-Medium sequence with a 1k transition speed, which consists of 7,500,000 images.

**Batch size.** We utilized two batch sizes: a standard value of 64, as in multiple previous works (Marsden et al., 2024; Press et al., 2023; Niu et al., 2022; Wang et al., 2021), and a lower one (equal to 10). Smaller batch sizes are more challenging in TTA because they result in an increased number of model updates and due to difficulties in batch statistics calculation. In some papers, even the batch size of 1 is used (Niu et al., 2023); in our experimental setup, we choose the less extreme value of 10, which is also commonly used in online continual learning (Mai et al., 2022). This means that for the longest sequence (CCC) each model is adapted 750,000 times.

**Methods**. We apply our modification to 4 teacher-student state-of-the-art frameworks: AdaContrast (Chen et al., 2022), CoTTA (Wang et al., 2022), RoTTA (Yuan et al., 2023a), PETAL (Brahma & Rai, 2023). Moreover, we report the performance with IT using (I-) prefix. Additionally, we com-

pare the results with the following state-of-the-art strategies that do not utilize teacher-student framework: TENT (Wang et al., 2021), EATA (Niu et al., 2022), SAR (Niu et al., 2023), RDUMB (Press et al., 2023) , and MEMO (Zhang et al., 2022). Non-adapted model is indicated as the Source and TestBN is a fixed model that uses batch normalization statistics from the current batch (Li et al., 2016; Schneider et al., 2020) as commonly done in TTA (Wang et al., 2021; 2022; Niu et al., 2022).. In-depth implementation details regarding the baselines are provided in the supplementary.

**Architectures.** Consistent with previous works (Wang et al., 2022; Marsden et al., 2024), we use WideResNet-28 (Zagoruyko & Komodakis, 2016) models with pre-trained weights from the *Robust-Bench* (Croce et al., 2021) model zoo for the main experiments on CIFAR10-C. Similarly, the tests on ImageNet-based benchmarks and DomainNet-126 employ the ResNet50 network with weights sourced from the same model zoo or those provided by (Marsden et al., 2024) for DomainNet-126.

Additional experiments in Section 6.2 are carried out with ResNet-26 GN (Wu & He, 2018), ResNeXt-50 (32x4d) (Xie et al., 2017), ViT-B16 (Dosovitskiy et al., 2021), and SwinViT-T (Liu et al., 2021) architectures. Weights for ResNet-26 GN are taken from (Zhang et al., 2022), as in (Marsden et al., 2024). *Torchvision* (maintainers & contributors, 2016) library is utilized to obtain the rest of the mentioned models.

**Implementation details.** As a testbed for experiments, we adopt the framework from Marsden et al. (2024). Experiments are conducted using parameters reported in the original papers. When running experiments on smaller batch sizes, we decrease the learning rate accordingly. We use parameters from standard experiments while testing on the long (L) scenarios.

## 6 EXPERIMENTAL RESULTS

Following the insights from the motivation experiment in Sec. 3, we evaluate the intransigent teacher modification over longer scenarios and compare the results with various state-of-the-art methods. Since this strategy extends existing methods, we refer to our proposed intransigent teacher with the prefix (I-) in front of the method being extended. All the proposed longer sequences than the usually reported ones in TTA are denoted with an (L) and consist of 20 loops over the commonly established adaptation scenarios. Further, the efficacy of the proposed approach is verified on numerous model architectures, including transformer-based ones. Finally, we present the evaluation of the hyperparameter selection robustness of our strategy.

### 6.1 ON ADAPTATION OVER LONG SCENARIOS

**The vulnerability of EMA teachers is revealed on extended test sequences.** In long scenarios (see Table 3), methods based on teacher-student framework often achieve lower performance compared to baselines or even cause the model to collapse, which is especially apparent on both ImageNet-C and CCC benchmarks. Those shortcomings are amplified in the lower batch size setting, potentially caused by a more significant error accumulation due to difficulties in estimating batch statistics and a larger number of adaptation steps. While the goal of using EMA is to provide a more stable adjustment and more accurate pseudo-labels for adaptation, we find this to not hold true for the longer settings. This is in contrast to the evaluation on common sequence length (see Supplemental Material), where the EMA teacher performs well and the original methods don't cause model collapses in most cases, except for the more challenging lower batch size setting. Results show that the performance of state-of-the-art TTA methods is clearly test sequence-length dependent.

**Intransigent teachers are very effective at collapse prevention.** The evaluated teacher-student adaptation methods exhibit some form of model collapse in 18 out of 30 cases for long sequences. When using an intransigent teacher, the collapse happens only twice for the small batch size. In these cases, performance is similar to the source model, yielding substantial improvements over the baseline - for instance, I-CoTTA achieves a 44.3% increase in accuracy on DomainNet (L).

**Intransigent teachers provide great reliability, out-of-the-box.** Keeping the teacher model intransigent on a batch size 64 allows for accuracy improvements of 12.2 (AdaContrast), 2.4 (CoTTA), and 8.0 (RoTTA) percentage points on average. It is even more effective on a smaller batch size, where the respective improvements are 22.8, 28.9, and 7.4. E.g., IT improved CoTTA on CIFAR10-C (L)

Table 3: Classification accuracy [%] for long scenarios. The value in superscript indicates the improvements over the baseline. **Bold** text indicates best performing method. Gray color indicates model collapse - performance worse than the non-adapting model (Source). Results averaged from 3 random seeds. * indicates the approximated result, details in the supplementary.

| Method | CIFAR10-C (L) | ImageNet-C (L) | ImageNet-R (L) | DomainNet-126 (L) | CCC | Avg. |
|---|---|---|---|---|---|---|
| Source | 56.5 | 18.0 | 36.2 | 54.7 | 16.8 | 36.4 |
| MEMO (Zhang et al., 2022) | 65.6 | 25.0 | 40.9 | 53.2 | 19.3* | 40.8 |
| BS = 10 | | | | | | |
| TestBN | 75.1 | 26.9 | 36.2 | 49.6 | 22.5 | 42.1 |
| TENT (Wang et al., 2021) | 39.0 | 4.7 | 17.4 | 10.9 | 0.7 | 14.5 |
| EATA (Niu et al., 2022) | 73.6 | 36.4 | **44.0** | 54.0 | **29.7** | 47.6 |
| SAR (Niu et al., 2023) | 75.2 | 30.6 | 43.6 | 50.8 | 20.3 | 44.1 |
| RDUMB (Press et al., 2023) | 76.8 | 34.3 | 40.1 | 51.5 | 28.1 | 46.2 |
| AdaContrast (Chen et al., 2022) | 72.1 | 2.3 | 8.0 | 47.0 | 0.4 | 26.0 |
| I-AdaContrast | **84.1**$^{+12.0}$ | **39.5**$^{+37.2}$ | 35.3$^{+27.3}$ | **63.2**$^{+16.2}$ | 21.8$^{+21.4}$ | **48.8**$^{+22.8}$ |
| CoTTA (Wang et al., 2022) | 23.8 | 3.8 | 33.7 | 6.0 | 17.3 | 16.9 |
| I-CoTTA | 69.7$^{+45.9}$ | 27.6$^{+23.8}$ | 37.4$^{+3.7}$ | 50.3$^{+44.3}$ | 25.7$^{+8.4}$ | 45.8$^{+28.9}$ |
| RoTTA (Yuan et al., 2023a) | 82.5 | 24.4 | 43.0 | 45.6 | 1.1 | 39.3 |
| I-RoTTA | 79.0$^{-3.5}$ | 33.6$^{+9.2}$ | 39.9$^{-3.1}$ | 57.8$^{+12.2}$ | 23.4$^{+22.3}$ | 46.7$^{+7.4}$ |
| PETAL (Brahma & Rai, 2023) | 67.9 | 2.4 | 36.6 | 49.6 | 0.8 | 31.5 |
| I-PETAL | 74.1$^{+6.2}$ | 26.6$^{+24.2}$ | 36.6$^{+0.0}$ | 49.5$^{-0.1}$ | 13.5$^{+12.7}$ | 40.6$^{+9.1}$ |
| BS = 64 | | | | | | |
| TestBN | 79.1 | 31.4 | 39.6 | 54.4 | 27.3 | 46.4 |
| TENT (Wang et al., 2021) | 20.1 | 11.1 | 36.4 | 18.4 | 1.2 | 17.4 |
| EATA (Niu et al., 2022) | 61.6 | 43.3 | 49.2 | 61.9 | 36.3 | 50.5 |
| SAR (Niu et al., 2023) | 79.2 | 39.9 | 47.3 | 59.2 | 22.3 | 49.6 |
| RDUMB (Press et al., 2023) | 81.1 | 41.7 | 47.5 | 59.0 | **37.0** | **53.3** |
| AdaContrast (Chen et al., 2022) | 81.8 | 18.8 | 26.5 | 61.7 | 2.4 | 38.2 |
| I-AdaContrast | **85.3**$^{+3.5}$ | 40.4$^{+21.6}$ | 38.1$^{+11.6}$ | **64.4**$^{+2.7}$ | 23.6$^{+21.2}$ | 50.4$^{+12.2}$ |
| CoTTA (Wang et al., 2022) | 56.0 | **52.8** | **50.5** | 45.6 | 8.3 | 42.7 |
| I-CoTTA | 68.3$^{+12.3}$ | 35.4$^{-17.4}$ | 39.5$^{-11.0}$ | 56.0$^{+10.4}$ | 26.3$^{+18.0}$ | 45.1$^{+2.4}$ |
| RoTTA (Yuan et al., 2023a) | 82.3 | 13.2 | 43.4 | 50.3 | 1.1 | 38.1 |
| I-RoTTA | 79.7$^{-2.6}$ | 32.9$^{+19.7}$ | 39.7$^{-3.7}$ | 56.6$^{+6.3}$ | 21.7$^{+20.6}$ | 46.1$^{+8.0}$ |
| PETAL (Brahma & Rai, 2023) | 58.3 | 31.5 | 39.7 | 54.5 | 16.0 | 40.0 |
| I-PETAL (Brahma & Rai, 2023) | 78.8$^{+20.5}$ | 31.2$^{-0.3}$ | 39.6$^{-0.1}$ | 54.5$^{+0.0}$ | 16.3$^{+0.3}$ | 44.1$^{+4.1}$ |

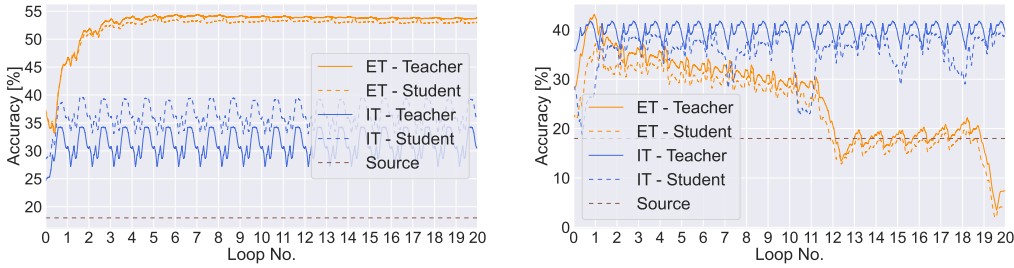

Figure 4: Per-batch accuracy on ImageNet-C (L) comparing CoTTA using ResNet50 **(left)** and ViT-B16 **(right)** models with EMA teacher (ET, orange) and intransigent teacher (IT, blue), both for teacher (solid) and student (dashed).

by 45.9% (from 23.8% to 69.7%). Note that applying this strategy did not require any hyper-parameter tuning nor any additional parameters. In fact, it removes the need to set the $\beta$ parameter.

**Intransigence can lead to slow adaptability.** There are certain situations in which using intransigent teacher might still lead to unreliable adaptation. Since it is a stability-based strategy, it lacks the capability to adapt to changes in the distribution rapidly or to explore the parameter space far away from the teacher model. This is especially visible for CoTTA on ImageNet-C (L) and ImageNet-R (L) when using a batch size of 64. In situations where it is possible to more accurately tune model hyper-parameters, using the EMA teacher can be favorable. It's important to note that CoTTA's exceptional performance on ImageNet-C relies on architectures with batch normalization. When applied to other architectures, it performs worse than its intransigent version (see Table 4).

**The student outperforming the teacher.** We show in Figures 1 and 4 the performance of both teachers and students for AdaContrast, CoTTA, and RoTTA, when we use either EMA or the intransigent teacher. In most cases, we can see that the student outperforms the teacher when the

intransigent teacher is used. The magnitude of this improvement is dependent on the used method, architecture, and dataset. When EMA is used, the teacher-student performance is (by design) tightly coupled. The CoTTA method performs significantly worse with ViT-B16 architecture. In this case, the student falls below the teacher but the IT keeps the student from degrading its performance further. More detailed results are presented in the Supplemental Material.

**On single image adaptation methods** Single image adaptation approaches, like MEMO (Zhang et al., 2022), are not susceptible to the studied issue of error accumulation. However, those methods have their problems, e.g., they are significantly more inefficient (Alfarra et al., 2024). Table A.1 from the supplementary material shows that MEMO is significantly slower than other techniques, with a wall-clock time around 20 times higher than the AdaContrast method. Moreover, it is outperformed by IT-modified methods and the baselines on most datasets, except ImageNet-R.

## 6.2 DETAILED ANALYSIS

**Intransigent teachers work across different architectures.** To verify if our findings hold for various model architectures, we present results on CIFAR10-C (L) and ImageNet-C (L) in Table 4. The learning rate is not tuned specifically for any of the models, and thus default values are used. The results with adjusted learning rates can be found in the supplementary (Table A.2). The IT is able to improve the TTA accuracy on long sequences for all of the compared models, including the ones without batch normalization layers. Therefore, rendering itself a highly universal approach.

**On robustness to learning rate selection.** In Table 5, we verify learning rate sensitivity by multiplying its value by 10 and 50. Default AdaContrast shows robustness when 10x learning rate is used, but collapses with a 50x learning rate on ImageNet-C. ITs allow increased robustness in all settings and achieve a non-collapsed solution, even for the highest learning rate. CoTTA lacks learning rate robustness, which is mitigated by the IT extension. Our strategy improves RoTTA's performance on CIFAR10-C. However, its low performance on ImageNet-C rendered our approach almost ineffective for this setup. Overall, ITs do not fully prevent the collapse when significantly altering the learning rates, although they seem to promote a better-performing parameter space.

**Effects varying temporal correlation.** Figure 5 illustrates the impact of varying degrees of class temporal correlation (Gong et al., 2022) on methods enhanced with IT. The analysis reveals that as temporal correlation increases, IT-enhanced methods consistently outperform their original versions, demonstrating superior robustness in these scenarios. This is the case even when the original method achieved better results on uniform class distribution (RoTTA). However, it should be noted that while IT-enhanced methods show improved performance, they are also negatively impacted by the effects of correlation. The degree of improvement varies significantly and is largely dependent on the underlying base technique employed.

**Are students better than intransigent teachers?** Figure 6 illustrates the accuracy difference between student and teacher models across various methods and datasets, allowing us to assess the student's learning capacity in relation to the intransigent teacher. In most cases, the student outperforms the teacher, with only two exceptions: I-AdaContrast on ImageNet-R (L) and I-CoTTA on CIFAR10-C (L). In these instances, the stability introduced by IT was insufficient to prevent degradation, only limiting further performance decline. Notably, after adjusting the learning rate (Figure 6 (right)), we observe no such cases of student underperformance.

Table 4: Classification accuracy [%] for long scenarios on CIFAR10-C and ImageNet-C with different model architectures. The value in superscript indicates the improvements over the baseline.

| | CIFAR10-C (L) | ImageNet-C (L) | | | |
|---|---|---|---|---|---|
| | ResNet26GN | ResNeXt-50 | ViT-B16 | SwinViT-T | ConvNeXt tiny |
| Source | 67.3 | 21.1 | 39.8 | 28.3 | 29.1 |
| AdaContrast (Chen et al., 2022) | 74.7 | 20.0 | 32.1 | 15.1 | 18.2 |
| I-AdaContrast | 79.4$^{+4.7}$ | 42.7$^{+22.7}$ | 43.5$^{+11.4}$ | 30.8$^{+15.7}$ | 32.5$^{+14.3}$ |
| CoTTA (Wang et al., 2022) | 16.7 | 57.0 | 26.2 | 0.1 | 0.1 |
| I-CoTTA | 61.6$^{+44.9}$ | 38.3$^{-18.7}$ | 30.9$^{+4.7}$ | 27.5$^{+27.4}$ | 16.3$^{+16.2}$ |
| RoTTA (Yuan et al., 2023a) | 66.0 | 16.2 | 36.2 | 7.3 | 18.7 |
| I-RoTTA | 72.5$^{+6.5}$ | 35.2$^{+19.0}$ | 42.9$^{+6.7}$ | 27.7$^{+20.4}$ | 29.7$^{+11.0}$ |

Table 5: Classification accuracy [%] for long scenarios on CIFAR10-C and ImageNet-C with 1x, 10x, 50x learning rate (LR) scaling. The superscript indicates the improvements over the baseline.

| | CIFAR10-C (L) | | | ImageNet-C (L) | | |
|---|---|---|---|---|---|---|
| | 1 x LR | 10 x LR | 50 x LR | 1 x LR | 10 x LR | 50 x LR |
| Source | | 56.5 | | | 18.0 | |
| AdaContrast (Chen et al., 2022) | 81.9 | 83.6 | 80.4 | 18.8 | 19.6 | 12.4 |
| I-AdaContrast | $85.4^{+3.5}$ | $86.1^{+2.5}$ | $85.2^{+4.8}$ | $40.4^{+21.6}$ | $36.3^{+16.7}$ | $32.7^{+20.3}$ |
| CoTTA (Wang et al., 2022) | 55.9 | 10.3 | 10.1 | 53.3 | 4.1 | 0.1 |
| I-CoTTA | $68.4^{+12.5}$ | $51.9^{+41.6}$ | $41.4^{+31.3}$ | $35.4^{-17.9}$ | $25.7^{+21.6}$ | $17.3^{+17.2}$ |
| RoTTA (Yuan et al., 2023a) | 82.3 | 80.6 | 28.9 | 13.8 | 0.2 | 0.1 |
| I-RoTTA | $79.6^{-2.7}$ | $77.9^{-2.7}$ | $70.6^{+41.7}$ | $32.7^{+18.9}$ | $4.1^{+3.9}$ | $2.1^{+2.0}$ |

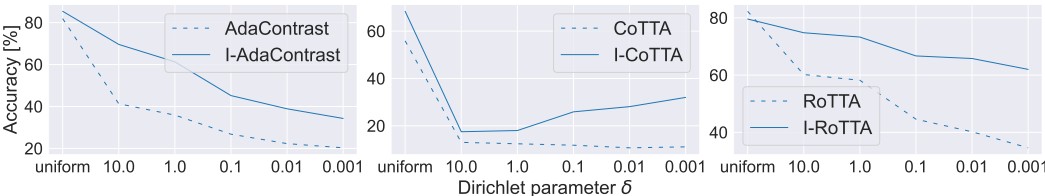

Figure 5: Classification accuracy on CIFAR10-C (L). Samples are sorted by class for different levels of correlation, by varying the Dirichlet concentration parameter $\delta$.

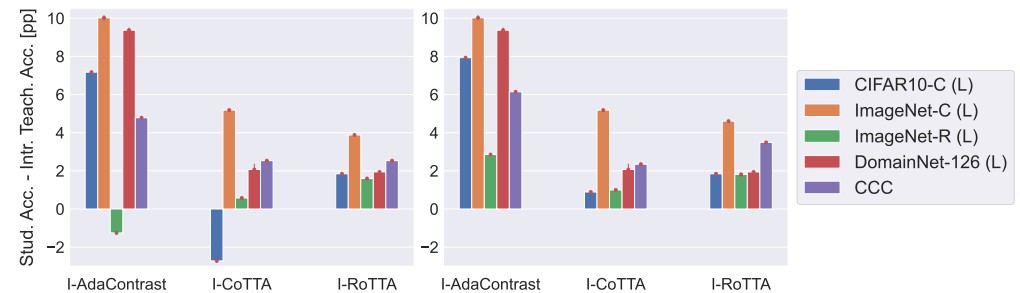

Figure 6: Accuracy difference between average student and intransigent teacher in percentage points, with default learning rate values (left) and after parameter search (right).

## 7 CONCLUSIONS

In this work, we explore the well-known strategy of using an exponential moving average teacher for test-time adaptation. We identify that this strategy, while being very effective at a common length of test sequences, has some shortcomings in longer ones, leading to model collapse on many benchmarks. After analyzing the plasticity-stability trade-off within existing teacher-student frameworks, we present a simple, effective strategy with an intransigent teacher that can be adapted to existing state-of-the-art methods. We show the efficacy of the presented modification by achieving significant improvements across many sequences, often preventing model collapse and increasing robustness to hyper-parameter changes. Most importantly, the benefits of this strategy are visible without any parameter tuning, it even removes the need to tune the $\beta$ parameter used in EMA teachers.

**Limitations.** It is unclear if the capabilities of the intransigent teacher become limited when there is some knowledge or prediction over the expected domain shift. However, the motivation behind the presented strategy is to provide consistent and reliable improvements when no assumptions about the distribution shift are available. Furthermore, we note that by adding the intransigent teacher to existing methods, we inherit their limitations. That is even though we improve the performance on many scenarios, if the baseline performs poorly, then using an intransigent teacher might not improve over the source (non-adapted model).

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

# A SUPPLEMENTARY

## A.1 DETAILS ON INTRANSIGENT TEACHER EXPERIMENT FROM SECTION 3

Our preliminary intransigent teacher experiment is conducted on ImageNet-C, a long ImageNet-C scenario (20x loop of standard ImageNet-C testing sequence), and CCC benchmarks. We utilize the same model as for our main experiments on ImageNet-based benchmarks - ResNet50 with pre-trained weights from the *RobustBench* (Croce et al., 2021) model zoo. We use a single loss within the teacher-student framework for the model adaptation during test time - either consistency loss from CoTTA (Consistency) or contrastive loss from AdaContrast (Contrastive). Any other components of the mentioned state-of-the-art methods are not included. Batch normalization statistics are recalculated for each batch. For both of the tested approaches, we use the SGD optimizer with a learning rate of 0.00025.

## A.2 BASELINES IMPLEMENTATION DETAILS

The experiments were conducted using the code repository of the previous test-time adaptation works (Marsden et al., 2024; Döbler et al., 2022). It provides the implementation of every tested state-of-the-art method. In terms of hyperparameters, we followed the implementations for tests on the typical batch size of 64.

TENT (Wang et al., 2021), EATA (Niu et al., 2022), SAR (Niu et al., 2023), and RDUMB (Press et al., 2023) use Adam optimizer with a learning rate of 0.001 for CIFAR10-C and SGD optimizer with a learning rate of 0.00025 for other benchmarks. AdaContrast (Chen et al., 2022) utilizes an SGD optimizer with a learning rate set to 0.0002 for all of the benchmarks. CoTTA (Wang et al., 2022) uses Adam optimizer with a learning rate of 0.001 for CIFAR10-C and SGD with a learning rate of 0.01 for the rest of the benchmarks. Adam optimizer with a learning rate set to 0.001 is used by RoTTA (Yuan et al., 2023a) for all of the tested datasets. MEMO (Zhang et al., 2022) uses an SGD optimizer with a learning rate of 0.005 for CIFAR10-C and 0.00025 for other datasets. PETAL (Brahma & Rai, 2023) in the original paper uses Adam optimizer with a learning rate of 0.001 for CIFAR10-C and SGD with a learning rate of 0.01 for other datasets. However, since we often experienced poor performance using these values on long scenarios, we utilized 10 times lower learning rates.

The learning rate used in experiments with batch size set to 10 was adjusted accordingly by scaling it linearly.

CoTTA (Wang et al., 2022) and PETAL (Brahma & Rai, 2023) methods update the student network using a consistency loss between the student and teacher. If the prediction confidence of the source model is below a certain threshold, the teacher's predictions are averaged over 32 different augmentations of the image which adds 31 additional forward operations of the neural network for each batch. It creates a significant computation overhead and causes the methods to be significantly slower, compared to other state-of-the-art methods. It is especially problematic for long adaptation sequence scenarios, which were the main part of our experiments. Our tests indicate that using a single augmentation does not alter the results notably. Therefore, for the ease of experimentation, we reduce the number of augmentations to 1.

The learning rate selection process for Figure 6 (right) was conducted using the Oracle method.

## A.3 DETAILS ON MEMO RESULTS ON CCC BENCHMARK FROM TABLE 3

The result is based on the first 623,000 images of the benchmark, providing an initial estimate of the method's accuracy. However, due to the benchmark's extensive size (7,500,000 images) and the method's requirement for a batch size of 1, we were unable to complete the full experiment in time. We estimate that processing the entire dataset will require approximately 972 hours on a single NVIDIA GeForce RTX 4080 GPU. This substantial time requirement underscores the method's significant computational inefficiency.

### A.4 COMPUTE DETAILS

All experiments were conducted on a single GPU. We utilized either NVIDIA A100 with 40GB of memory or NVIDIA GeForce RTX 4080 with 16GB of memory. Execution time of experiment greatly varied and was dependent on the dataset, scenario (standard or long), tested method and batch size. The fastest experiments took about 30 minutes, whereas the longest lasted up to 36 hours.

### A.5 DISCUSSION ON COTTA AND I-COTTA PERFORMANCE ON IMAGENET-C (L) AND IMAGENET-R (L)

I-CoTTA underperforms compared to the original CoTTA on ImageNet-C (L) and ImageNet-R (L) with a batch size of 64 and architectures with batch normalization layers, as shown in Table 3 and Table 4. The accuracy drops by 17.4 and 10.8 percentage points, respectively. We attribute this to CoTTA's exceptional performance in these specific scenarios, where it outperforms all other tested methods and achieves a stable performance improvement as presented in Figure 4 (left). The additional regularization from IT doesn't enhance stability in this case. Instead, it over-regularizes the student model, hindering its adaptation capability. This case, while unusual for CoTTA (considering other CoTTA results), demonstrates that IT isn't universally effective. However, it's crucial to note that even in this case, IT still outperforms the source model. Our focus is on improving the overall reliability of TTA across all settings, not just in specific scenarios where certain methods may excel. Also, note that COTTA does not perform that well on architectures without batch normalization layers.

### A.6 WALL-TIME RESULTS

Table A.1: The wall-clock time (seconds) for processing 10,000 images of CIFAR10C on a single RTX 4080 GPU.

| Method | Time [s] |
|---|---|
| Source | 3.4 |
| MEMO | 508.4 |
| AdaContrast | 25.3 |
| I-AdaContrast | 25.0 |
| CoTTA | 40.7 |
| I-CoTTA | 40.2 |
| RoTTA | 27.7 |
| I-RoTTA | 27.5 |

### A.7 RESULTS WITH DIFFERENT ARCHITECTURES AND LEARNING RATES

Table A.2 presents additional results using different neural network architectures. The learning rate was tuned by the Oracle method to provide favorable conditions for the original TTA approaches and ensure they work correctly. All results from the learning rate selection process are in Table A.3. The intransigent teacher is able to improve the test-time adaptation accuracy on long sequences for all of the compared models even when the original methods have tuned learning rates specifically for tested sequence length.

Table A.2: Classification accuracy [%] for long scenarios on CIFAR10-C and ImageNet-C with different neural network architectures. The value in superscript indicates the improvements over the baseline. The learning rate parameter is adjusted using the Oracle method. The batch size is equal to 64.

| | CIFAR10-C (L) | ImageNet-C (L) | | | |
| | ResNet26GN | ResNeXt-50 | ViT-B16 | SwinViT-T | ConvNeXt tiny |
|---|---|---|---|---|---|
| Source | 67.3 | 21.1 | 39.8 | 28.3 | 29.1 |
| AdaContrast (Chen et al., 2022) | 75.7 | 38.8 | 41.5 | 30.6 | 33.4 |
| I-AdaContrast | $79.6^{+3.9}$ | $42.7^{+3.9}$ | $43.5^{+2.0}$ | $30.9^{+0.3}$ | $32.5^{-0.9}$ |
| CoTTA (Wang et al., 2022) | 57.0 | 42.1 | 41.7 | 28.4 | 29.1 |
| I-CoTTA | $67.3^{+10.3}$ | $38.3^{-3.8}$ | $40.7^{-1.0}$ | $28.9^{+0.5}$ | $30.5^{+1.4}$ |
| RoTTA (Yuan et al., 2023a) | 70.2 | 35.6 | 40.6 | 28.8 | 29.0 |
| I-RoTTA | $72.5^{+2.3}$ | $36.2^{+0.6}$ | $42.9^{+2.3}$ | $28.9^{+0.1}$ | $29.7^{+0.7}$ |

Table A.3: Classification accuracy [%] for long scenarios on CIFAR10-C and ImageNet-C with different neural network architectures and learning rates with the batch size equal to 64. Intransingent versions are much more robust to changes in hyperparameters.

| | LR | CIFAR10-C (L) | ImageNet-C (L) | | | |
| | | ResNet26GN | ResNeXt-50 | ViT-B16 | SwinViT-T | ConvNeXt tiny |
|---|---|---|---|---|---|---|
| Source | - | 67.3 | 21.1 | 39.8 | 28.3 | 29.1 |
| AdaContrast | 0.001 | **75.7** | 20.0 | 29.6 | 13.0 | 17.5 |
| | 0.0002 | 74.7 | 20.0 | 32.1 | 15.1 | 18.2 |
| | 0.00025 | 75.1 | 20.3 | 31.8 | 14.4 | 18.2 |
| | 3.125e-5 | 74.3 | 25.6 | 39.0 | 21.9 | 22.4 |
| | 1e-6 | 72.0 | **38.8** | **41.5** | 29.6 | 32.0 |
| | 1e-7 | 68.4 | 33.5 | 40.7 | **30.6** | **33.4** |
| | 1e-8 | 68.1 | 32.1 | 40.0 | 28.7 | 31.2 |
| I-AdaContrast | 0.001 | **79.6** | 39.5 | 42.1 | 30.4 | 32.4 |
| | 0.0002 | 79.4 | **42.7** | **43.5** | 30.8 | **32.5** |
| | 0.00025 | 79.5 | 42.4 | 43.4 | 30.8 | **32.5** |
| | 3.125e-5 | 77.7 | 42.3 | 43.1 | **30.9** | 32.3 |
| | 1e-6 | 73.0 | 37.6 | 42.0 | **30.9** | 31.7 |
| | 1e-7 | 69.2 | 33.4 | 41.0 | 30.3 | 30.5 |
| CoTTA | 0.01 | 12.3 | **57.1** | 26.2 | 0.1 | 0.1 |
| | 0.001 | 16.6 | 42.1 | 34.5 | 26.3 | 0.2 |
| | 0.00025 | 14.8 | 39.2 | 38.7 | 25.5 | 19.3 |
| | 3.125e-5 | 26.6 | 39.3 | **41.7** | **28.4** | 22.2 |
| | 1e-6 | 56.2 | 33.7 | 40.0 | 27.0 | **29.1** |
| | 1e-7 | **57.0** | 33.0 | 39.4 | 28.0 | 29.0 |
| | 1e-8 | **57.0** | 32.9 | 39.3 | 28.2 | 29.1 |
| I-CoTTA | 0.01 | 26.9 | 38.3 | 30.9 | 27.5 | 16.3 |
| | 0.001 | 61.7 | 35.9 | 39.9 | **28.9** | 27.6 |
| | 0.00025 | 62.1 | **36.0** | 40.1 | 28.7 | 29.3 |
| | 3.125e-5 | 62.1 | 35.7 | **40.7** | 28.6 | 29.8 |
| | 1e-6 | **67.3** | 33.6 | 40.4 | 28.3 | **30.5** |
| | 1e-7 | **67.3** | 33.0 | 39.9 | 28.3 | 28.3 |
| RoTTA | 0.001 | 66.0 | 16.2 | 36.2 | 7.3 | 18.7 |
| | 0.00025 | 68.5 | 19.7 | 34.8 | 7.9 | 16.8 |
| | 3.125e-5 | **70.2** | **35.6** | 36.5 | 13.9 | 16.3 |
| | 1e-6 | 68.5 | 33.6 | **40.6** | **28.8** | 26.8 |
| | 1e-7 | 67.9 | 31.2 | 40.0 | 28.4 | **29.0** |
| | 1e-8 | 67.3 | 30.8 | 39.8 | 28.3 | **29.0** |
| I-RoTTA | 0.001 | **72.5** | 35.2 | **42.9** | 27.7 | **29.7** |
| | 0.00025 | 72.4 | **36.2** | 42.6 | 27.3 | **29.7** |
| | 3.125e-5 | 71.2 | 34.3 | 42.1 | 27.7 | 29.0 |
| | 1e-6 | 68.9 | 28.3 | 40.7 | **28.9** | 28.5 |
| | 1e-7 | 67.9 | 26.1 | 40.0 | 28.4 | 29.0 |

## A.8 EFFECTS OF INTRANSIGENCE AMOUNT EXTENDED EXPERIMENT

To signify the point of Section 4.2, Figure A.1 shows results where the test sequence was extended to 100 loops of common CIFAR10-C. It verified that CoTTA with ET and $\beta = 0.9999$ degrades below the performance of IT, given enough samples in the test sequence. This observation highlights a significant issue of TTA methods, as they can face test sequences of arbitrary lengths after deployment.

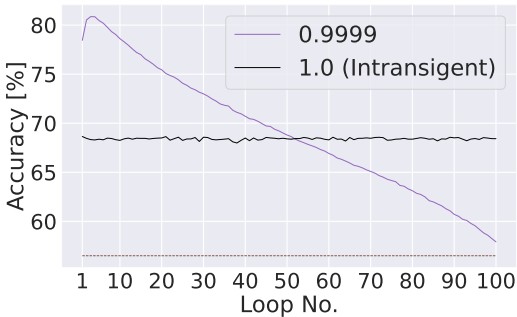

Figure A.1: Mean accuracy [%] of CoTTA with varying $\beta$ for each loop of common CIFAR10-C testing sequence repeated 100 times. The brown dashed line indicates the accuracy of the source model as a reference.

## A.9 TUNING LEARNING RATE VALUE FOR LONG SCENARIOS.

We investigated whether tuning the learning rate, arguably the most crucial hyperparameter, could enhance the performance of baseline methods in long adaptation scenarios. Following a realistic approach, we employed an Oracle technique on ImageNet-C (L) as a reference benchmark (inspired by Rusak et al. (2022), we call it Transfer IN-C) and applied the selected learning rate across all datasets. The results, presented in Table A.4, reveal the complexity of hyperparameter optimization in test-time adaptation.

Our findings shows the challenges of hyperparameter tuning. For instance, CoTTA achieved superior accuracy with its default learning rate compared to the tuned version. While AdaContrast and RoTTA showed improvements with optimized learning rates, our IT approach consistently outperformed these methods, even when they were specifically tuned for long-sequence adaptation. These results underscore both the difficulty of hyperparameter selection and the robust performance of our IT method across varying conditions.

Table A.4: Classification accuracy [%] for long scenarios with the learning rate (LR) parameter tuned. LR value **Default** means that the default LR value for the method was used. **Transfer IN-C** indicates that the LR is tuned utilizing the ImageNet-C benchmark with ground truth labels. The batch size is equal to 64.

| Method | LR value | CIFAR10-C (L) | ImageNet-C (L) | ImageNet-R (L) | DomainNet-126 (L) | Avg. |
|---|---|---|---|---|---|---|
| AdaContrast | Default | 81.8 | 18.8 | 26.5 | 61.7 | 47.2 |
|  | Transfer IN-C | 81.2 | 36.1 | 40.8 | 59.7 | 54.5 |
| I-AdaContrast | Default | 85.4 | 40.4 | 38.2 | 64.4 | 57.1[+9.9] |
|  | Transfer IN-C | 85.4 | 40.4 | 38.2 | 64.4 | 57.1[+2.6] |
| CoTTA | Default | 56.0 | 52.8 | 50.5 | 45.6 | 51.2 |
|  | Transfer IN-C | 11.2 | 52.8 | 50.5 | 45.6 | 40.0 |
| I-CoTTA | Default | 68.4 | 35.4 | 39.6 | 56.8 | 50.1[-1.1] |
|  | Transfer IN-C | 52.0 | 35.4 | 39.6 | 56.8 | 46.0[+6.0] |
| RoTTA | Default | 82.3 | 13.2 | 43.4 | 50.3 | 47.3 |
|  | Transfer IN-C | 73.2 | 30.8 | 41.0 | 55.3 | 50.1 |
| I-RoTTA | Default | 79.6 | 32.7 | 39.7 | 57.2 | 52.3[+5.0] |
|  | Transfer IN-C | 79.3 | 33.3 | 39.9 | 57.2 | 52.4[+2.3] |

## A.10  POTENTIAL OF ADAPTIVE $\beta$ VALUE.

In Table A.5, we explore a dynamic approach to adjusting the teacher model's momentum parameter ($\beta$). Our experiment begins with the default value of $\beta = 0.999$, allowing initial teacher model plasticity, then transitions to complete weight preservation of IT ($\beta = 1.0$) after one full cycle through the data. This hybrid approach outperforms our IT technique in several cases, demonstrating the potential of adaptive momentum strategies.

However, the results are not uniformly positive with our standard IT outperforming the hybrid method in some cases (AdaContrast on ImageNet-C (L) and CoTTA on DomainNet-126 (L)). This suggests that the fixed period length is not a universal value and there is a need to adjust it correctly.

Table A.5: Classification accuracy [%] for long scenarios with the weights of the teacher fixed only after the 1st loop on the test sequence. The value in superscript indicates the improvements over the IT technique's performance. The batch size is equal to 64.

| Method | CIFAR10-C (L) | ImageNet-C (L) | ImageNet-R (L) | DomainNet-126 (L) | Avg. |
|---|---|---|---|---|---|
| AdaContrast | $85.2^{-0.1}$ | $38.4^{-2.0}$ | $38.2^{+0.1}$ | $65.3^{+0.9}$ | $56.8^{-0.3}$ |
| CoTTA | $72.0^{+3.7}$ | $45.0^{+9.6}$ | $42.8^{+3.3}$ | $49.1^{-6.9}$ | $52.2^{+2.4}$ |
| RoTTA | $80.4^{+0.7}$ | $36.1^{+3.2}$ | $41.0^{+1.3}$ | $57.9^{+1.3}$ | $53.9^{+1.7}$ |

## A.11  DISCUSSION ON MODEL RESET MECHANISM.

CoTTA's proposed resetting mechanism aims to preserve source knowledge by stochastically restoring portions of the student model's weights to their original source state during each update iteration. In principle, an effective source knowledge preservation technique should eliminate the need for our IT technique.

However, CoTTA's reset mechanism introduces a restoration probability parameter. To ensure our findings were not biased by suboptimal parameter selection, we conducted parameter tuning experiments, documented in Table A.6. These results reveal that the optimal restoration probability varies across datasets, with model performance dependent on this parameter. When following a realistic scenario of tuning on a single dataset, the performance improvements were marginal (Avg. Transfer IN-C). Only by using an Oracle approach on all benchmarks, we observe performance gains, highlighting the practical limitations of this approach.

Table A.6: Classification accuracy [%] for long scenarios with restoration probability parameter $p$ of CoTTA method tuned. The batch size is equal to 64. **Avg. Def.** is the average accuracy with default $p$ value. **Avg. Transfer IN-C** is the average accuracy with a single $p$ value chosen on the ImageNet-C dataset using the Oracle method. Average accuracy when the $p$ value is chosen separately for each of the datasets with Oracle is presented in **Avg. Oracle** column.

| $p$ value | CIFAR10-C (L) | ImageNet-C (L) | ImageNet-R (L) | DomainNet-126 (L) | Def. | Avg. Transfer IN-C | Oracle |
|---|---|---|---|---|---|---|---|
| 0.1 | 73.1 | 29.0 | 41.8 | 26.9 | | | |
| 0.01 | 53.7 | 24.8 | 35.6 | 13.7 | | | |
| 0.001 (Def.) | **56.0** | 52.8 | **50.5** | 45.6 | 51.2 | 51.6 | 58.1 |
| 0.0001 | 54.7 | **53.7** | 45.0 | 52.9 | | | |
| 0.00001 | 54.3 | 53.6 | 49.0 | **55.0** | | | |
| 0.0 | 52.7 | 53.5 | 48.9 | 54.5 | | | |

## A.12  DISCUSSION ON RDUMB.

RDumb has already been established as a state-of-the-art baseline method for extended adaptation scenarios, demonstrating great performance in both prior work (Press et al., 2023) and our current experiments. Despite its effectiveness, limitations should be considered.

The method's mechanism of periodically resetting the model to its initial state leads to significant accuracy drops immediately following each reset, as illustrated in Figure A.2. Such instability is particularly concerning since reliable test-time adaptation should maintain consistent performance

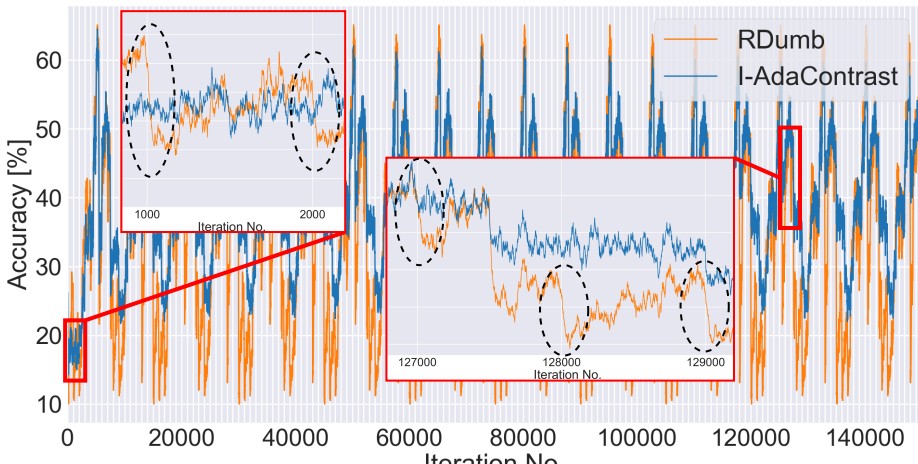

Figure A.2: Batchwise accuracy plots of RDumb and I-AdaContrast methods on ImageNet-C (L) benchmark. The accuracy values were smoothed to make the plot clearer. RDumb resets the model every 1000 iterations, which causes significant drops in accuracy after the reset.

throughout the adaptation process. Furthermore, the same constant reset interval is likely not optimal for every case, which adds a hyperparameter to select. In contrast, our IT approach achieves comparable performance without requiring parameter tuning.

### A.13 ADAPTATION TO REPEATED SOURCE DOMAIN DATA.

We investigated whether the observed accuracy degradation during adaptation stems solely from distribution shift by conducting experiments on the source domain's validation splits. We evaluated performance under two conditions: a single pass through the data (1x) and 20 repeated passes (20x), with results shown in Table A.7. Our findings reveal that accuracy degradation occurs even on source domain data, with dataset-specific variations. This phenomenon is visible on all tested datasets except CIFAR10-C. We attribute this exception to CIFAR10-C's lower complexity, particularly its smaller number of classes compared to other datasets in our study.

The IT in most cases improves the performance on repeated streams (20x), however, the increased stability negatively impacts the accuracy on the 1x streams (especially with CoTTA and RoTTA).

Table A.7: Classification accuracy [%] for the adaptation on the source domain's validation splits. 1x indicates the performance on a single pass through the data, while 20x means the accuracy on the 20 repeated passes. The batch size is equal to 64. The degradation of performance also occurs when adapting to the source domain, however, this effect depends on the dataset and the method used.

| Method | CIFAR10-C | | ImageNet-C | | ImageNet-R | | DomainNet-126 | | Avg. | |
|---|---|---|---|---|---|---|---|---|---|---|
| | 1x | 20x | 1x | 20x | 1x | 20x | 1x | 20x | 1x | 20x |
| AdaContrast | 93.6 | 93.7 | 72.3 | 38.4 | 91.4 | 87.1 | 93.2 | 85.7 | 87.6 | 76.2 |
| I-AdaContrast | 93.6 | 93.7 | 72.8 | 66.5 | 91.4 | 88.8 | 94.1 | 92.8 | 88.0 | 85.5 |
| CoTTA | 93.5 | 92.9 | 74.2 | 63.2 | 91.7 | 90.2 | 86.1 | 61.2 | 86.4 | 76.9 |
| I-CoTTA | 77.4 | 81.6 | 51.0 | 60.5 | 77.5 | 88.1 | 74.1 | 84.7 | 70.0 | 78.7 |
| RoTTA | 94.2 | 94.4 | 75.7 | 63.2 | 91.9 | 81.5 | 89.1 | 58.8 | 87.7 | 74.5 |
| I-RoTTA | 94.1 | 93.5 | 73.1 | 72.9 | 90.7 | 85.4 | 68.8 | 88.0 | 81.7 | 85.0 |

### A.14 ADDITIONAL RESULTS

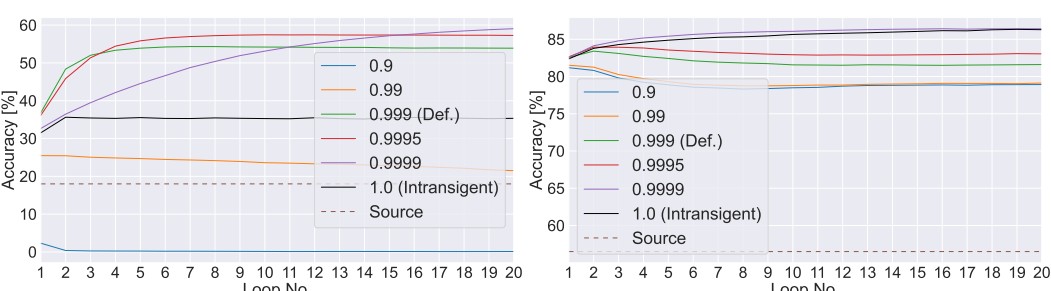

Figure A.3: Mean accuracy [%] for each loop of common testing sequence on ImageNet-C (L) using CoTTA **(left)** and on CIFAR10-C (L) using AdaContrast **(right)**. The Brown dashed line indicates the Source model accuracy.

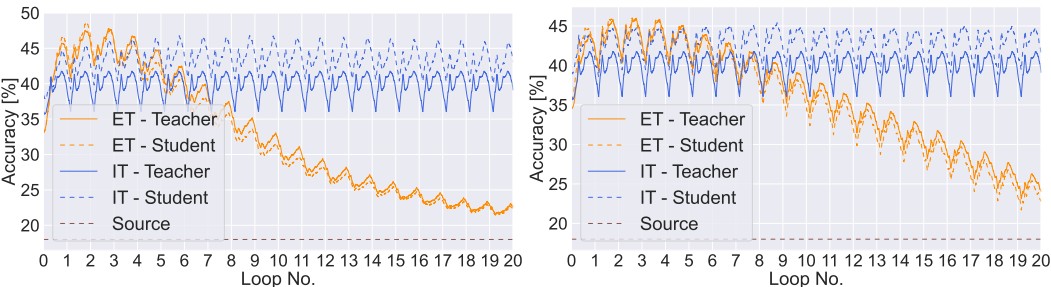

Figure A.4: Per batch accuracy [%] on ImageNet-C (L) comparing AdaContrast **(left)** and RoTTA **(right)** using ViT-B16 network with EMA teacher (ET, orange) and intransigent teacher (IT, blue), both for teacher (solid) and student (dashed).

Table A.8: Classification accuracy [%] for common length sequences.

| Method | CIFAR10-C | ImageNet-C | ImageNet-R | DomainNet-126 |
|---|---|---|---|---|
| Source | 56.5 | 18.0 | 36.2 | 54.7 |
| MEMO (Zhang et al., 2022) | 65.6 | 25.0 | 40.9 | 53.2 |
| BS = 10 | | | | |
| TestBN | 75.0 | 27.0 | 36.6 | 46.5 |
| TENT (Wang et al., 2021) | 75.7 | 31.2 | 38.9 | 52.4 |
| EATA (Niu et al., 2022) | 77.4 | 36.0 | 43.1 | 54.4 |
| SAR (Niu et al., 2023) | 75.8 | 31.3 | 41.9 | 52.8 |
| RDUMB (Press et al., 2023) | 77.2 | 34.8 | 41.3 | 52.0 |
| AdaContrast (Chen et al., 2022) | 81.3 | 33.3 | 39.5 | 56.5 |
| I-AdaContrast | 82.0 | 33.8 | 39.8 | 59.6 |
| CoTTA (Wang et al., 2022) | 75.1 | 26.4 | 41.1 | 52.0 |
| I-CoTTA | 69.8 | 28.3 | 35.6 | 49.5 |
| RoTTA (Yuan et al., 2023a) | 79.0 | 29.2 | 38.6 | 55.9 |
| I-RoTTA | 73.2 | 29.4 | 39.3 | 56.6 |
| PETAL (Brahma & Rai, 2023) | 68.3 | 23.2 | 36.6 | 49.5 |
| I-PETAL | 74.2 | 27.3 | 36.6 | 49.5 |
| BS = 64 | | | | |
| TestBN | 79.2 | 31.4 | 39.7 | 54.5 |
| TENT (Wang et al., 2021) | 77.8 | 37.3 | 42.6 | 58.0 |
| EATA (Niu et al., 2022) | 79.8 | 42.0 | 45.8 | 59.7 |
| SAR (Niu et al., 2023) | 79.3 | 37.8 | 42.8 | 57.2 |
| RDUMB (Press et al., 2023) | 81.4 | 40.0 | 46.2 | 58.9 |
| AdaContrast (Chen et al., 2022) | 82.6 | 34.8 | 40.9 | 62.0 |
| I-AdaContrast | 82.4 | 35.1 | 41.0 | 61.7 |
| CoTTA (Wang et al., 2022) | 82.2 | 36.8 | 42.8 | 58.9 |
| I-CoTTA | 68.6 | 31.7 | 35.9 | 54.4 |
| RoTTA (Yuan et al., 2023a) | 80.9 | 32.4 | 39.2 | 56.8 |
| I-RoTTA | 76.7 | 30.6 | 39.3 | 56.3 |
| PETAL (Brahma & Rai, 2023) | 76.6 | 31.5 | 39.7 | 54.5 |
| I-PETAL | 78.4 | 31.4 | 39.7 | 54.5 |

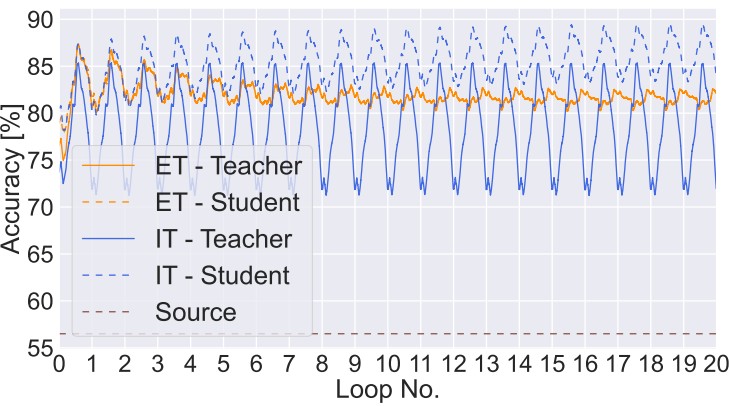

Figure A.5: Per batch accuracy [%] on CIFAR10-C (L) using AdaContrast and WideResNet-28 network with EMA teacher (ET, orange) and intransigent teacher (IT, blue), both for teacher (solid) and student (dashed).

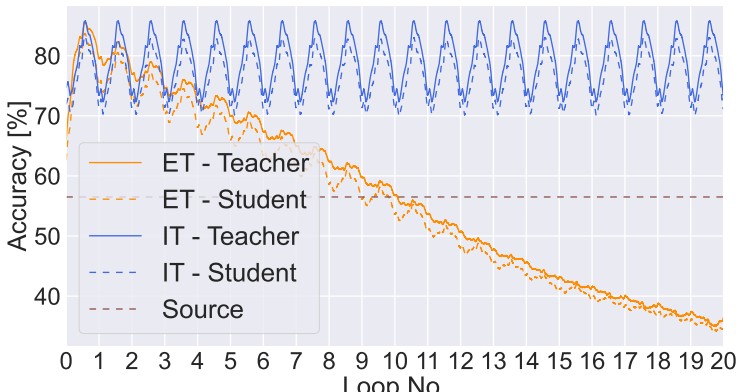

Figure A.6: Per batch accuracy [%] on CIFAR10-C (L) using CoTTA and WideResNet-28 network with EMA teacher (ET, orange) and intransigent teacher (IT, blue), both for teacher (solid) and student (dashed).

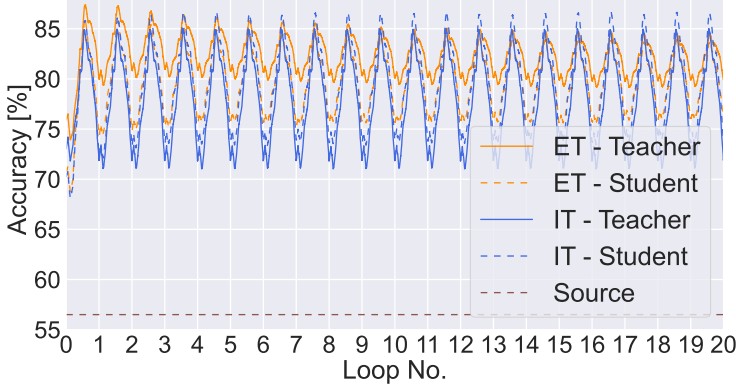

Figure A.7: Per batch accuracy [%] on CIFAR10-C (L) using RoTTA and WideResNet-28 network with EMA teacher (ET, orange) and intransigent teacher (IT, blue), both for teacher (solid) and student (dashed).

