# OpenReview forum: "Intransigent Teachers Guide Better Test-Time Adaptation Students"
_ICLR.cc/2025/Conference — Submitted to ICLR 2025_

### Official Review · Reviewer_3dWc · 2024-10-21

**Soundness:** 3
**Presentation:** 4
**Contribution:** 2
**Rating:** 5
**Confidence:** 5

**Summary:**

The paper challenges the existing teacher-student framework in test-time adaptation (TTA), where lifelong adaptations showed inevitable performance degradation. The paper proposes an intransigent teacher, which does not update the parameters but only uses test batch statistics. The intransigent teacher showed high stability in lifelong adaptation while improving performance compared to the original teacher-student-based methods.

**Strengths:**

- Writing is clear, comprehensive, and easy to understand.

- Proposed a simple yet effective solution for the practical scenario of lifelong adaptation.

- Extensive large-scale evaluations on various datasets/scenarios and state-of-the-art baselines.

**Weaknesses:**

- The problem (model failures in lifelong adaptation) has already been discussed in RDumb, so the problem setting itself is not novel.

- The method only applies to existing teacher-student methods, thus limiting its applicability. At the same time, the intransigent teacher does not consistently outperform the baselines (e.g., RDumb in BS=64) or prevent failures (e.g., results in BS=10).

**Questions:**

- Please discuss the advantages/disadvantages of the proposed intransigent teacher compared to the important lifelong baseline, RDumb.

- Can we dynamically adjust the plasticity ($\beta$) to climb up to 1 (e.g., using the TTA accuracy estimation metrics [a, b] or using a fixed period)?

- Would this phenomenon also occur in non-corrupted lifelong test streams?

- Reporting single-pass results (akin to the original TTA setup) would help understand the performance compared to existing TTAs.

- (Minor) Typo: Page 8, Line 423: COTTA -> CoTTA

---

[a] Lee, Taeckyung, et al. "AETTA: Label-Free Accuracy Estimation for Test-Time Adaptation." Proceedings of the IEEE/CVF Conference on Computer Vision and Pattern Recognition. 2024.

[b] Kim, Eungyeup, et al. "Reliable Test-Time Adaptation via Agreement-on-the-Line." arXiv preprint arXiv:2310.04941. 2024.

---

> ### Author Response · Authors · 2024-11-18
> **Clarification**
>
> Many thanks for the insightful review. We are happy that you noticed the effectiveness of the proposed technique, the extensiveness of our experiments, and described our writing as clear and comprehensive.
> Thank you for pointing out the typo. Before we fully respond, could you please provide below clarification to let us fully understand your review?
>
> > _""Would this phenomenon also occur in non-corrupted lifelong test streams?"_
>
> Could you please elaborate on what streams do you have in mind? Our experiments include non-corrupted streams (ImageNet-R, DomainNet).

---

> > ### Comment · Reviewer_3dWc · 2024-11-18
> >
> > I wondered whether the phenomenon (model collapse and intransigent teachers) would occur in a normal test stream (e.g., lifelong ImageNet test stream without corruption) to check whether this phenomenon occurs due to test-time distribution shifts or lifelong evaluation. I would appreciate any discussions (not necessarily asking for the experiment).

---

> > > ### Author Response · Authors · 2024-11-26
> > >
> > > Thank you for the clarification. We would like to address the entirety of the comments.
> > >
> > > **W1.**
> > >
> > > See global response A.
> > >
> > > Similar to us, RDumb focuses on long scenarios and provides a straightforward analysis. However, there are several key differences between our work and RDumb.
> > >
> > > First, RDumb experiments exclusively with common corruptions, while we demonstrate the phenomenon across various types of distribution shifts, including ImageNet-R and DomainNet. Additionally, we analyze this issue within several newer methods that incorporate the mean-teacher mechanism, which theoretically should address the problem (whereas only CoTTA among RDumb’s baselines used this mechanism). Furthermore, we highlight a potential, simple solution to mitigate the issue, which is not explored in RDumb.
> > >
> > > **W2.**
> > >
> > > See global response D.
> > >
> > > Our primary contribution is not the introduction of a novel method that outperforms existing SOTA approaches, but rather highlighting a significant performance issue in current TTA methods.
> > >
> > > We demonstrate the effectiveness of the Intransigent Teacher (IT) technique in addressing the specific challenge of performance degradation on longer test sequences. While IT may occasionally underperform, the performance gap is typically small. For instance, CoTTA shows strong performance with ResNet (Table 3), but this does not generalize well to other architectures (Table 4) or altered hyperparameters (Table 5), where it significantly underperforms compared to the I-CoTTA variant.
> > >
> > > The IT approach provides two key advantages:
> > >
> > > - Increased reliability: the IT strategy reduces instances where adaptation performs worse than no adaptation at all, which serves as the ultimate baseline for TTA methods.
> > > - Simplified hyperparameter selection: this approach addresses a major challenge in TTA by simplifying hyperparameter tuning.
> > >
> > > Although rare failure cases exist, we believe the insights and performance analysis presented here will be valuable to the community and encourage further research in this area.
> > >
> > > **Q1.**
> > >
> > > Firstly, a clear disadvantage in RDumb's resetting the adapted model to the initial state causes the sudden accuracy drops on batches directly after that, which we now show in Figure A.2. We argue that a reliable TTA method should provide as stable performance as possible. Moreover, the same reset interval is likely not optimal for every case, therefore it requires additional hyperparameters to tune. IT does not require any parameter tuning.
> > >
> > > **Q2.**
> > >
> > > See global response C.
> > >
> > > Utilizing the TTA accuracy estimation metrics to adjust the plasticity seems like a promising approach, but we did not manage to proceed with that idea experimentally due to time constraints.
> > >
> > > **Q3.**
> > >
> > > As requested by the reviewer, we performed such experiments, by adapting to (clean) validation sets of selected datasets (Table A.7). The results suggests that this phenomenon indeed occur in some lifelong evaluation settings, even on the source domain data. The issue seems to be the most visible on ImageNet-C, ImageNet-R and DomainNet-126 datasets. We hypothesize that this might be the result of the higher difficulty of the datasets considering significantly higher number of classes, compared to the number of classes in CIFAR10-C dataset. Therefore, test-time distribution shifts might not be the most important factor for the model collapse to happen during the long adaptation process.
> > >
> > > The IT tends to improve the performance on repeated non-corrupted streams (20x), however, the increased stability negatively impact the accuracy on the 1x streams (especially with CoTTA and RoTTA).
> > >
> > > **Q4.**
> > >
> > > Single-pass results are presented in the supplementary material (Table A.3). To summarize briefly, the accuracy of the default baselines and IT modifications is generally comparable, though IT is often slightly outperformed when using a batch size of 64. It’s important to note that the default baselines have undergone parameter tuning, whereas no parameter adjustments were made for the IT modifications. Conversely, IT's increased stability becomes more advantageous with smaller batch sizes and a higher number of updates.
> > >
> > > While IT can occasionally result in lower performance on standard-length sequences, it is crucial to consider that in real-world deployments, the length of the adaptation sequence is unpredictable.
> > >
> > > Although IT may not always achieve the highest performance across all sequence lengths, it consistently delivers reliable results across all scenarios in our experimental settings. This contrasts with other methods, such as AdaContrast, which perform exceptionally well on standard-length sequences but fail to maintain performance on the extended sequences we propose.
> > >
> > > ---
> > > We trust that our explanations have addressed the reviewer's concerns. Should there be any additional questions, we are more than willing to provide further details. If no further clarification is needed, we kindly ask the reviewer to reconsider the final score.

---

> > > > ### Comment · Reviewer_3dWc · 2024-11-26
> > > >
> > > > I appreciate the authors' thorough rebuttal with additional experiments.
> > > >
> > > > However, I still have concerns about accuracy drops with IT in ImageNet-C/R experiments. Considering ImageNet-C is an important benchmark, the accuracy drop demonstrates that IT might not generally apply. CoTTA (with stochastic reset) shows that applying IT reduces the accuracy by 17.4%/11.0%, questioning the necessity of IT.
> > > >
> > > > Also, considering this paper does not present a new TTA method, the novelty must rely on new findings and their impact. However, I am still concerned about the (1) similar observations in RDumb (although not explored in IN-R and DomainNet or some new methods, as the authors stated in the rebuttal) and (2) many TTA methods are not based on teacher-student frameworks.
> > > >
> > > > Therefore, I decided to maintain the rating. I will keep track of the discussions and update if the above concerns are addressed.

---

> > > > > ### Author Response · Authors · 2024-11-26
> > > > >
> > > > > We appreciate the reviewer’s engagement in the discussion and detailed follow-up comments.
> > > > >
> > > > > > _"However, I still have concerns about accuracy drops with IT in ImageNet-C/R experiments. Considering ImageNet-C is an important benchmark, the accuracy drop demonstrates that IT might not generally apply. CoTTA (with stochastic reset) shows that applying IT reduces the accuracy by 17.4%/11.0%, questioning the necessity of IT."_
> > > > >
> > > > >
> > > > > **1. ResNet-50 specificity of CoTTA**: We want to emphasize that the extraordinary long-sequence accuracy achieved by CoTTA is specific to the ResNet-50 architecture. In contrast, CoTTA's performance on ImageNet-C for other architectures (ViT-B16, SwinViT-T and ConvNeXt, Table A.2) is comparable to the source model. Importantly, achieving these results often requires careful hyperparameter tuning (Tables 4, 5, and A.3). In this context, IT offers competitive accuracy while being significantly less sensitive to hyperparameter changes.
> > > > >
> > > > > **2. Accuracy trade-offs with IT**: While IT presents a drop in accuracy comparable to the baseline method on ImageNet-C/R with ResNet-50, it still significantly outperforms the source model in most cases. IT is designed as a conservative, robust approach that prioritizes stability. Although its relative gains may be smaller when the baseline achieves substantial improvements over the source model, its performance is closely tied to the teacher's, significantly reducing the probability of falling below the source model. Moreover, IT provides increased resilience to hyperparameter changes. Both of these qualities —minimizing the risk of underperformance and ensuring robustness to hyperparameters— are critical for real-world applications.
> > > > >
> > > > > **3. Novelty of the observation**: Finally, we would like to stress that IT being significantly more robust is not the main point anyway, but rather the fact that such a simple yet robust method could solve the issue in many scenarios, even if not universally.
> > > > >
> > > > > > _"Also, considering this paper does not present a new TTA method, the novelty must rely on new findings and their impact. However, I am still concerned about the (1) similar observations in RDumb (although not explored in IN-R and DomainNet or some new methods, as the authors stated in the rebuttal) and (2) many TTA methods are not based on teacher-student frameworks."_
> > > > >
> > > > > We firmly believe this work has the potential to make a significant impact on the test-time adaptation (TTA) community.
> > > > >
> > > > > **1. Teacher-Student Frameworks**: There is an increasing number of methods utilizing teacher-student frameworks without a thorough understanding of their behavior or limitations. By demonstrating the surprising effect that using a fixed teacher can be an effective adaptation technique — allowing the student to significantly outperform the teacher (e.g., Figures 4, 6, A.4) — we anticipate that this will inspire future research to:
> > > > >
> > > > > **a)** Develop new methods specifically leveraging fixed teachers, or
> > > > >
> > > > > **b)** Enhance the flexibility of the IT approach by allowing dynamic teacher updates - IT could potentially adapt to new data over a limited number of initial steps (as suggested by the reviewer and now explored by us in Table A.5)), or
> > > > >
> > > > > **c)** Promote broader analysis of the proposed methods that are not limited to the specific benchmarks, but also explore their usability in more realistic, lifelong scenarios.
> > > > >
> > > > > **2. Alternative to RDumb**: We also emphasize that extending the array of experiments provides valuable insights. For instance, our results highlight that even RDumb has its failure modes (e.g., DomainNet with smaller batch sizes). While IT is inherently a conservative approach, RDumb focuses on rapid adaptation and uses a reset mechanism via the ETA method. Our work presents an alternative approach for long-term adaptation scenarios, which does not rely on reset mechanisms.

---

> > > > > > ### Comment · Reviewer_3dWc · 2024-11-28
> > > > > >
> > > > > > Thank you for your detailed response and clarification.
> > > > > >
> > > > > > I want to clarify that when I referenced 'similar observations in RDumb,' I specifically referred to the observed TTA failures in lifelong adaptation, not the reset-based methodology. These observations about model instability during lifelong adaptation align with existing understanding in the field.
> > > > > >
> > > > > > While your work makes a contribution, I believe there may be opportunities to strengthen its impact on the TTA community. While effective, the use of a fixed model as a stabilizing factor through regularization follows somewhat naturally from existing teacher-student approaches.
> > > > > >
> > > > > > I think your work could be particularly valuable if you expanded the investigation of IT. For example, what was the reasoning behind basing IT on BatchNorm statistics rather than fixing it? Such analysis could provide deeper insights into adaptation mechanism design.
> > > > > >
> > > > > > After careful consideration of your response, I will maintain my original score.

---

> > > > > > > ### Author Response · Authors · 2024-12-01
> > > > > > >
> > > > > > > > _"I want to clarify that when I referenced 'similar observations in RDumb,' I specifically referred to the observed TTA failures in lifelong adaptation, not the reset-based methodology. These observations about model instability during lifelong adaptation align with existing understanding in the field."_
> > > > > > >
> > > > > > > It is true that RDUMB also observed the TTA failures in lifelong adaptation. Yet, we believe that our current contributions (as described in the introduction) still stay strong:
> > > > > > >
> > > > > > > a) Analysis of the EMA teacher framework in TTA.
> > > > > > >
> > > > > > > b) Observation that a fixed teacher (IT) can guide surprisingly strong students (e.g., Fig. 1 and Fig. 6).
> > > > > > >
> > > > > > > c) Extensive empirical evaluation.
> > > > > > >
> > > > > > > > _"I think your work could be particularly valuable if you expanded the investigation of IT. For example, what was the reasoning behind basing IT on BatchNorm statistics rather than fixing it? Such analysis could provide deeper insights into adaptation mechanism design."_
> > > > > > >
> > > > > > > Thank you for the interesting question. Using batch normalization (BN) statistics computed at test time is a common strategy in TTA. The simplest approach, known as the TestBN method, involves calculating new BN statistics for each batch at test time and serves as a solid baseline (Table 3). In our case, we adopted the same strategy for computing BN statistics as the base method. CoTTA utilizes precisely the TestBN technique. In terms of AdaContrast and RoTTA, they update BN statistics using exponential moving average-based approaches.
> > > > > > >
> > > > > > > To clarify the above points we have also run the experiments with frozen teacher's statistics at test time which were calculated during training on source data (batch size is set to 64):
> > > > > > >
> > > > > > > | | CIFAR10-C (L) | ImageNet-C (L) | ImageNet-R (L) | DomainNet-126 (L) | Avg.
> > > > > > > | -------- | -------- | -------- | -------- | -------- | -------- |
> > > > > > > | I-AdaContrast | 83.2 | 24.8 | 35.3 | 63.3 | 51.7$^{-5.4}$ |
> > > > > > > | I-CoTTA | 49.4 | 17.6 | 35.5 | 52.3 | 38.7$^{-11.1}$ |
> > > > > > > | I-RoTTA | 61.1 | 16.2 | 37.4 | 51.8 | 41.6$^{-10.6}$ |
> > > > > > >
> > > > > > > The value in superscript indicates a decline in performance compared to the IT technique’s performance with updated BN statistics. Results suggest that adjusting the BN statistics in the teacher model indeed improves overall performance. Those results align with those of the TTA community, which indicate that correcting the statistics can significantly improve performance on out-of-distribution data.
> > > > > > >
> > > > > > > Please note that the above relates only to CNN architectures with BN. In (Tables 4, A.2, and A.3) we also experiment with other architectures without BN layers, e.g., transformers.

---

### Official Review · Reviewer_3kuq · 2024-11-03

**Soundness:** 1
**Presentation:** 2
**Contribution:** 1
**Rating:** 3
**Confidence:** 5

**Summary:**

An interesting phenomenon found in the paper is that the intransigent teacher model is able to guide a more stable student model in long sequences of CTTA tasks. From the conclusions in the paper, it is clear that this approach can be applied to all methods of the mean-teacher architecture.

**Strengths:**

The phenomenon observed in the paper does to some extent replace the current method of the mean-teacher architecture and can achieve better results in long sequences of CTTA tasks.

**Weaknesses:**

The text lacks critical experimental and theoretical proofs and does not present targeted methods and analyses.

Limitations：
1. The paper does not provide a detailed analysis, but is only based on experimentally observed phenomena, and it is not possible to determine the specific reasons for the decline in generalizability of the teacher model, nor does it give a specific analysis of the decline in generalizability performance of the teacher model.
2. Why the intransigent teacher model outperforms the EMA updated teacher model in the long sequence CTTA task, relying only on experimental comparisons is not convincing.
3. Is the Intransigent teacher model just setting β to 1? How is this different from freezing the model? Is it understood to always use the source model as the teacher model? If so, it is no longer considered to be a mean-teacher framework.
4. Should the teacher model be locked in any scenario? It is suggested that the authors consider a scenario where the weights of the teacher model are dynamically adjusted, which might achieve better results.
5. Based on the phenomena you observed, the paper doesn't seem to suggest any targeted approach? Does this imply that you are just using the source model as a teacher model? I don't see any relevant methods in the source code either.
6. Although comparisons were made on three methods in the paper, the paper should have added more comparison experiments with the Mean-Teacher architecture method. Also, the authors need to provide segmentation experiments to further demonstrate the effectiveness of their proposed approach.

**Questions:**

See the weakness.

---

> ### Author Response · Authors · 2024-11-18
> **Clarification**
>
> We thank the reviewer for the valuable feedback. Before we fully respond, we would like to ask you to provide below clarification to help us better understand your review.
>
> > _"6. [...] Also, the authors need to provide split experiments to further demonstrate the effectiveness of their proposed approach."_
>
> Could you also please clarify what you mean by 'split experiments'?

---

> > ### Comment · Reviewer_3kuq · 2024-11-25
> >
> > It means the segmentation experiments.

---

> ### Author Response · Authors · 2024-11-26
>
> Thank you for the clarification.
>
> We have received overall good feedback about our presentation quality from the other reviewers, including an excellent score from reviewer *3dWc*. However, we are concerned about the lower score of 2 from the reviewer’s assessment. To help us further improve the presentation, could the reviewer kindly provide more specific guidance on which sections or aspects require refinement?
>
> Below we answer to the remaining weaknesses pointed by the reviewer:
>
> > _"The text lacks critical experimental and theoretical proofs and does not present targeted methods and analyses."_
>
> We selected three state-of-the-art (SOTA) methods that utilize the Mean-Teacher framework, as they represent well-established benchmarks in this domain. Additionally, we now present results for the PETAL method (Tables 3 and A.8) mentioned by reviewer *aKAL*.
>
> Using the default learning rates provided in the PETAL paper resulted in poor performance. After conducting a few preliminary runs, we opted to reduce the default learning rates by a factor of 10, which yielded better performance. The average improvement achieved by applying the intransigent teacher (IT) was 9.1% and 4.1% for batch sizes of 10 and 64, respectively.
>
> The PETAL method appears to be largely robust on ImageNet-R (L) and DomainNet-126 (L), demonstrating similar accuracy on both long and default scenarios. However, on CIFAR10-C (L) (for batch size 64) and ImageNet-C (L), performance decreases in longer scenarios compared to the default ones. In these cases, IT enhances PETAL's performance, mitigating the issue. Additionally, IT improves accuracy on the CCC benchmark.
>
> **1.**
>
> See global response B.
>
> We believe that the teacher-student framework does not eliminate error accumulation in TTA but simply delays it. This is evident in Figures 2 and 4, where we show that in the standard EMA framework, the accuracy of the teacher and the student tends to converge, with the teacher’s accuracy eventually aligning with that of the student after a delay. Additionally, we demonstrate that even using a very high β parameter value of 0.9999 does not prevent performance collapse, as shown in Figure A1, where adaptation is tested over a sequence repeated 100 times.
>
> **2.**
>
> See global response A.
>
> We provide an observation of an issue and a very simple technique to mitigate that. We do not say that IT outperforms EMA updated teacher in every case. This is purely experimental observation, which is also presented experimentally.
>
> **3.**
>
> See global response A.
>
> That’s correct. The approach is equivalent to freezing the model, except when batch normalization layers are present. In such cases, the batch normalization statistics are updated in the same manner as in the base method. The reviewer is also correct that, strictly speaking, this would no longer fall under the mean-teacher framework. For clarity, we referred to EMA-based teachers as "mean-teachers" and to fixed teachers as "intransigent teachers" to ensure a clear distinction in our presentation.
>
> **4.**
>
> See global response C.
>
> In practice, if we can make some assumption about the distribution shift or adaptation sequence length, adjusting the β parameter, instead of setting it to 1, could work better (in some scenarios). In this work we do not make any of such assumptions.
>
> **5.**
>
> See global response A.
>
> We believe that the observation of the phenomena is important to understand current methods in their context, relevant for the community when exploring further realistic cases, highlighting that a simple method, which is a specific hyperparameter case for some existing approaches, seems to be able to navigate the issue better than more complex specialized methods under the proposed scenario.
>
> **6.**
>
> As discussed above, we have now added results with the PETAL architecture.
> Due to the timing of this response, we were unable to include results for semantic segmentation, as this would require identifying and adapting a new codebase. Nevertheless, the current results across various baseline methods, architectures, datasets, and scenario types demonstrate the robustness of using a fixed teacher.
>
> ---
> We hope our explanation alleviates any concerns the reviewer may have. Should there be any additional inquiries, we are more than willing to provide further details. If no further clarification is needed, we kindly ask the reviewer to reconsider the final score.

---

> > ### Comment · Reviewer_3kuq · 2024-11-27
> >
> > Thanks to the response, after confirming more specific details about the paper, I still have the following concerns:
> >
> > 1.  Based on the authors' response, it is entirely feasible to use the outputs of the student model during training as pseudo-labels in this setup. I believe that "Intransigent Teachers" is no longer based on the Mean-Teacher framework if the teacher model is not used.
> > 2. The paper does not provide a specific theoretical analysis or methodological design. Even if the experimental findings are useful, the paper does not offer enough guidance for future work, making it hard to push the research forward.
> > 3. The paper lacks more experiments to further validate the findings, and the fact that "Intransigent Teachers" is not superior to EMA in all scenarios shows that the simple solution proposed in the paper is still flawed to some extent.
> >
> > Therefore, I believe the current manuscript is still incomplete, and I decide to maintain the rating.

---

> > > ### Author Response · Authors · 2024-12-01
> > >
> > > > 1. _"Based on the authors' response, it is entirely feasible to use the outputs of the student model during training as pseudo-labels in this setup. I believe that "Intransigent Teachers" is no longer based on the Mean-Teacher framework if the teacher model is not used."_
> > >
> > > Yes, as answered in our previous response (**3.**) the IT is, strictly speaking, no longer a mean-teacher approach.
> > > We would like to thank the reviewer for an interesting experiment questioning the use of the teacher-student framework in TTA, by removing the teacher. We have run a set of experiments, using only the student network. Considering the lack of regularization and weight averaging of the teacher, we verified multiple learning rate values to obtain reasonable results. The result for each learning rate is presented in the table:
> > >
> > > | Using only student model | LR | CIFAR10-C (L) | ImageNet-C (L) | ImageNet-R (L) | DomainNet-126 (L) |
> > > | -------- | -------- | -------- | -------- | -------- | -------- |
> > > | AdaContrast | 1e-5 | **82.4** | 23.0 | 37.4 | 59.1 | 54.7 | 54.5 |
> > > | AdaContrast | 1e-6 | 81.4 | **36.6** | **40.7** | **59.2** | | |
> > > | AdaContrast | 1e-7 | 79.1 | 31.9 | 39.6 | 57.0 | | |
> > > | -------- | -------- | -------- | -------- | -------- | -------- | -------- | -------- |
> > > | CoTTA | 1e-4 | 35.3 | 6.3 | 14.1 | 1.1 | 40.6 | 37.1 |
> > > | CoTTA | 1e-5 | 70.8 | **34.8** | **41.3** | 1.6 | | |
> > > | CoTTA | 1e-6 | **79.2** | 31.9 | 40.0 | **7.0** | | |
> > > | -------- | -------- | -------- | -------- | -------- | -------- | -------- | -------- |
> > > | RoTTA | 1e-4 | 21.9 | 1.4 | 5.8 | 6.9 | 45.9 | 45.9 |
> > > | RoTTA | 1e-5 | 51.6 | 14.5 | 35.6 | 48.9 |  | |
> > > | RoTTA | 1e-6 | **86.6** | **26.0** | **37.3** | **53.8** |  | |
> > >
> > > The table below summarizes the results, averaging the best achieved accuracies (Oracle Avg.), the accuracies with the learning rate chosen on ImageNet-C (L) benchmark (Transfer IN-C Avg.) and accuracies with the learning rate chosen on standard unrepeated benchmarks (Transfer 1xLoop Avg.). The value in superscript indicates the decline of performance compared to the IT technique’s performance.
> > > | Using only student model | Oracle Avg. | Transfer IN-C Avg. |  Transfer 1xLoop Avg. |
> > > | -------- | -------- | -------- | -------- |
> > > | AdaContrast | 54.7$^{-3.6}$ | 54.5$^{-2.6}$ | 50.5$^{-6.6}$ |
> > > | CoTTA | 40.6$^{-12.5}$ | 37.1$^{-8.9}$ | 26.7$^{-23.1}$ |
> > > | RoTTA | 45.9$^{-6.5}$ | 45.9$^{-6.5}$ | 41.4$^{-10.8}$ |
> > >
> > > Using only the student results in inferior performance across all experiments compared to the IT. However, it can perform reasonably well if hyperparameters close to optimal are found, though results generally exhibit greater variance.
> > >
> > > > 2. _"The paper does not provide a specific theoretical analysis or methodological design. Even if the experimental findings are useful, the paper does not offer enough guidance for future work, making it hard to push the research forward."_
> > >
> > > While we do not provide theoretical justification, we give solid empirical evidence instead. We believe that the work can have a significant impact on the field. Future work could use the findings of our paper by:
> > >
> > > a) developing new methods specifically leveraging fixed teachers,
> > >
> > > b) enhancing the flexibility of the IT approach by allowing dynamic teacher updates - IT could potentially adapt to new data over a limited number of initial steps (as explored in Table A.5),
> > >
> > > c) promoting broader analysis of the new methods that are not limited to the specific benchmarks, but also explore their usability in more realistic, lifelong scenarios.
> > >
> > > > 3. _"The paper lacks more experiments to further validate the findings, and the fact that "Intransigent Teachers" is not superior to EMA in all scenarios shows that the simple solution proposed in the paper is still flawed to some extent."_
> > >
> > > We welcome suggestions for additional experiments that could further enhance our study.
> > > Our paper provides the most extensive benchmarking of long sequences to date, covering five datasets (Table 3), various architectures (Tables 4, A.2, A.3), correlated data streams (Fig. 5), and different hyperparameter selection strategies (Tables A.2-A.4 and A.6).
> > > IT was applied to four baseline methods, demonstrating significant improvements in average accuracy, substantial prevention of model collapse, and emphasizing the need for novel methods that perform better over longer sequences.
> > > As noted in our global response A, we acknowledge the simplicity and limitations of IT, but our goal is *not* to introduce a state-of-the-art method.

---

### Official Review · Reviewer_iaLD · 2024-11-03

**Soundness:** 3
**Presentation:** 3
**Contribution:** 3
**Rating:** 8
**Confidence:** 4

**Summary:**

This paper addresses the challenges of Test-Time Adaptation (TTA) and proposes to use a static (intransigent) teacher model, which does not update its weights during adaptation. The authors demonstrate that this modification enhances performance across multiple datasets characterized by longer sequences and smaller batch sizes. Additionally, they provide evidence that their proposed method is adaptable across various model architectures and exhibits robustness against changes in hyper-parameters.

**Strengths:**

1.  The "intransigent teacher" concept is a fresh perspective that challenges existing methodologies in TTA, potentially leading to improved performance in real-world applications, such as LLM applications.
2. The authors support their claims with experimental results across multiple datasets, demonstrating the effectiveness of their approach in diverse scenarios. The proposed method shows robustness to hyper-parameter variations.
3. The proposed approach is simple and can be generalized across different architectures.

**Weaknesses:**

1. While the empirical results are compelling, the theoretical justification for why the intransigent teacher improves performance could be elaborated further to enhance the understanding of the underlying mechanisms.
2. The implications of using an unchanging teacher model over extended periods or across highly variable data distributions could be discussed more thoroughly, as this might lead to stagnation in learning.

**Questions:**

See the weaknesses for details.

---

> ### Author Response · Authors · 2024-11-26
>
> We are grateful to the reviewer for the positive assessment of our work. We appreciate the recognition that our work offers a novel perspective that challenges existing TTA methodologies. We respond to the weaknesses pointed out in the review:
>
> **1.**
>
> > _"While the empirical results are compelling, the theoretical justification for why the intransigent teacher improves performance could be elaborated further to enhance the understanding of the underlying mechanisms."_
>
> See global response B.
>
> **2.**
>
> >_"The implications of using an unchanging teacher model over extended periods or across highly variable data distributions could be discussed more thoroughly, as this might lead to stagnation in learning."_
>
> This is an interesting point. We hypothesize that stagnation probably depends on the combination of both dataset and utilized method. Figure 3 shows that the performance of AdaContrast with IT on ImageNet-C (L) (left, black) is improved over the first four loops. However, the accuracy of CoTTA with IT on CIFAR10-C (L) (right, black) stagnates. We believe that the stagnation is not necessarily a bad effect, considering the performance is better than that of the source model.
>
> ---
>
> If the reviewer's concerns have been sufficiently addressed in our responses, we humbly seek the support of our paper towards acceptance. If there are any further concerns or additional points to raise, we are eager to address them.

---

### Official Review · Reviewer_aKAL · 2024-11-04

**Soundness:** 2
**Presentation:** 3
**Contribution:** 1
**Rating:** 5
**Confidence:** 3

**Summary:**

The paper proposes an intransigent teacher (IT) based approach for continual test-time adaptation (TTA), where the teacher model is kept frozen, and only the student model updates. The aim is to alleviate the problem of error accumulation that is persistent in longer horizons of target domains.
Experimental results on longer horizons of corruption sequences demonstrate that IT helps improve performance in compared settings on multiple benchmarks.

**Strengths:**

* Experiment with different approaches that use losses, such as consistency and contrastive losses.
* Improving performance on longer horizons on multiple benchmarks

**Weaknesses:**

* Limited novelty. EMA-based continual TTA approaches already have a hyperparameter that decides how much weightage to be given to the student model weights and updates the teacher weights as a linear combination. If the weightage to student model weights is extremely low, it is effectively an "intransigent teacher."
* CoTTA [1] and PETAL [2] have already proposed a resetting mechanism that preserves source knowledge by resetting some weights back to the source pre-trained model.
* Repeated loops of the same data showing poor performance can also mean that the model is overfitting to each target domain and drifting away from source knowledge, which is suitable for all the target domains. Approaches such as CoTTA [1] and PETAL [2] have a resetting mechanism that consists of a threshold hyperparameter while resetting. Tuning this hyperparameter is essential for longer horizons using the validation corruption data. Otherwise, the comparison with baselines is not fair.
* The proposed approach is limited to EMA student-teacher models.

**Questions:**

* If we refer to CoTTA paper [1] Equation 2 and its supplementary [2], \alpha (\beta in the submitted paper) can be put to 1, and it will effectively lead to an "intransigent teacher." Is this understanding correct? If so, what is the novelty of this paper, and why is it not just a trivial extension in terms of methodology?
* Is the paper simply not setting the \beta value to 1 and experiments around it?
* Is repeating the same corruption sequence multiple times realistic? If the paper claims intransigent teacher helps, there should be new benchmarks with longer horizons of corruption sequences, rather than repeating the corruption sequence.
* Tuning this hyperparameter is essential for longer horizons using the validation corruption data. Otherwise, the comparison with baselines is not fair. Have the authors tuned the hyperparameters for the baseline approaches? Also, was any validation corruption data used?

**References**
1. Qin Wang, Olga Fink, Luc Van Gool, and Dengxin Dai. Continual test-time domain adaptation. In Proceedings of the IEEE/CVF Conference on Computer Vision and Pattern Recognition, 2022
2. https://openaccess.thecvf.com/content/CVPR2022/supplemental/Wang_Continual_Test-Time_Domain_CVPR_2022_supplemental.pdf
3. Dhanajit Brahma, and Piyush Rai. A probabilistic framework for lifelong test-time adaptation. Proceedings of the IEEE/CVF Conference on Computer Vision and Pattern Recognition. 2023.

---

> ### Author Response · Authors · 2024-11-26
>
> Many thanks for the valuable review. We would like to address some of the comments and answer the questions raised.
>
> **W1**
>
> See global response A.
>
> The novelty of the approach does not lie in considering the IT as an "option" by tuning the corresponding hyperparameter. The key point is that previous work did not compare with it due to the limited scope of the testing sequence. In the scenarios we consider, the analysis changes drastically. We do show that the IT becomes a somewhat trivial solution given the problem we highlight. Therefore, the novelty is in the analysis of such methods. And the importance for future work to be aware of such baselines is also relevant.
>
> **W2**
>
> That’s correct; however, they do not fully mitigate the issue, as demonstrated in our work on CoTTA. For the rebuttal, we have also included results with the PETAL method (see Tables 3 and A.8).
>
> Using the optimal reset mechanism requires careful hyperparameter tuning, particularly for long adaptation scenarios. We tuned the reset parameter of the CoTTA method, as detailed in Table A.6. The results are presented for three scenarios: using the default parameter value (0.01, as applied in all experiments from the original CoTTA paper), using Oracle selection, and using the optimal parameter determined based on the ImageNet-C (L) scenario, inspired by the approach of Rusak et al. [R2].
>
> The findings indicate that the optimal parameter value varies across datasets, significantly impacting final accuracy. Moreover, introducing resetting mechanisms inherently adds hyperparameters, which are not straightforward to tune. This poses a challenge for real-world applications where the test data distribution is unknown, making it impractical to rely on a similar hold-out set for hyperparameter selection.
>
> That said, resetting mechanisms show promise for adaptation in extended scenarios. Fixed-teacher in an alternative approach, which offers some benefits as discussed now in the supplementary (Section A.12).
>
> **W3, Q4**
>
> We argue that the assumption of correctly tuning the hyperparameters is overoptimistic and assumes that the validation data is similar to the test data (in terms of both domain shift and the length of the validation sequence).
>
> In the main paper, we utilized the optimal hyperparameters provided by the original authors and did not perform any tuning for the IT approach, highlighting its inherent robustness. As requested by the reviewer, we conducted additional experiments, tuning the reset parameter for CoTTA (Table A.6) and learning rates for all teacher-student-based methods (Table A.4). For these experiments, we adopted the Oracle hyperparameter selection technique on ImageNet-C (L), inspired by [R2], and applied the selected parameters to other datasets.
>
> This hyperparameter selection strategy did not yield significantly better results compared to using the default values, as observed in both the reset parameter and learning rate tuning experiments. While tuning the reset parameter resulted in slightly improved accuracy, the learning rate experiments produced mixed outcomes, with some cases even showing decreased accuracy due to the chosen selection method. Only the unrealistic scenario of applying the Oracle technique to each dataset individually noticeably improved the average accuracy in the reset experiments.
>
> We further emphasize the general robustness of the IT approach to hyperparameter changes, as demonstrated in Table A.3.
>
> **W4**
>
> See global response D.
>
> **Q1, Q2**
>
> See global response A.
>
> That is correct. We have outlined our contributions in the general response. They include showing that the common technique of using EMA teacher is simply not ideal in TTA.
>
> **Q3**
>
> Our objective was to evaluate performance on significantly longer sequences. However, the only available benchmark we found for this purpose was the CCC benchmark, which we used in our experiments. Recognizing that a single corruption-based benchmark is insufficient, we proposed our repeated sequence benchmark. It is important to note that corruptions are not the sole type of domain shift we consider, as evidenced by our experiments on the ImageNet-R and DomainNet-126 datasets.
>
> We acknowledge that the proposed repeated sequence scenario offers limited variability. However, we argue that if existing methods struggle in this controlled setting (as our results demonstrate), they are unlikely to perform well in more variable or complex real-world scenarios. Additionally, we observe consistent results on the CCC benchmark, which does not rely on repeated sequences, further supporting our findings.
>
> ---
> We hope our explanation alleviates any concerns the reviewer may have.  Should there be any additional queries, we are more than willing to provide further details. If no further clarification is needed, we kindly ask the reviewer to reconsider the final score.
>
> ---
>
> [R2] Rusak et al. "If your data distribution shifts, use self-learning.", TMLR 2022.

---

> > ### Author Response · Authors · 2024-12-01
> >
> > Dear Reviewer aKAL,
> >
> > The end of the discussion period is approaching.
> > We have made every effort to address your concerns through additional experiments (e.g., detailed parameter selection, incorporating a reset mechanism, and adding the PETAL method) and by providing further clarifications.
> >
> > The reviewer comments allowed us to improve the quality of our work and we hope these updates align with your expectations and address your concerns. We'd be happy to engage further during the remaining discussion period if there are any remaining issues or additional feedback you would like us to consider.

---

### Author Response · Authors · 2024-11-26

First and foremost, we would like to thank all the reviewers for the insightful feedback on our work. We are particularly encouraged that reviewers *iaLD* and *3dWc* recognized the simplicity and effectiveness of our solution to the described problem. Additionally, reviewers *iaLD* and *3kuq* highlighted the broad applicability of our technique across various architectures and test-time adaptation methods based on the mean-teacher framework.

We have made the following changes to the revised version:
- Experiments with the new method (PETAL) with and without the Intransigent Teacher (Tables 3, A.8). The average improvement achieved by applying the intransigent teacher (IT) was 9.1% and 4.1% for batch sizes of 10 and 64, respectively.
- Results with parameter tuning (i.e., for baseline methods, Tables A.3, A.4, A.6),
- Results with increased plasticity by allowing the teacher to change for an initial fixed number of steps (Table A.5),
- Comparison with RDumb method (discussion and Fig. A.2).
- Results on source domain data streams (Table A.7).

Changes in text are indicated by the violet font color in the revised pdf version.

The results presented above highlight the robustness of using a fixed teacher, particularly in relation to variations in hyperparameters and corruption types.

We welcome further discussions on any aspects of the paper that may require additional clarification. In particular, there are four recurring concerns raised by the reviewers, which we would like to address collectively:

**A. Limited Novelty. Method too simple, is it just setting β to 1.0?**

To make it absolutely clear, it is in fact setting the β value in the linear combination of weights to 1. This means that the weights of the teacher in the mean-teacher framework become fixed. Output predictions for the evaluation are taken from the student model, regardless of the method to which the IT is applied. One relevant detail is that we also update the statistics in batch normalization layers of the teacher model in the way the respective original base method did if the corresponding model has such layers.

We identify a significant limitation in current TTA methods, specifically their performance degradation on unusually long test sequences. This observation is valuable for the community as it highlights a crucial flaw in common evaluation protocols and SOTA methods utilizing the teacher-student framework.
In response to this issue, we offer a straightforward solution: the use of the IT. While simple, this approach effectively mitigates the described problem. Our primary goal is NOT to introduce a novel method that outperforms all existing SOTA approaches. Rather, we aim to:

1. Draw attention to the performance issue in current TTA methods,
2. Investigate commonly used teacher-student framework and its shortcomings,
3. Address the issues with a simple method, even though it is sometimes outperformed by other SOTA methods.

The novelty of our work lies in identifying the performance of degradation issue on extended test sequences and proposing a simple yet effective solution to mitigate this specific problem within the widely utilized teacher-student framework.

**B. Lack of theoretical justification.**

While our current study lacks a comprehensive theoretical framework, we present comprehensive experimental evidence documenting this previously unreported phenomenon. Our goal is to bring attention to these findings and establish a good starting point for future research as we have not yet uncovered the underlying theoretical justifications. We welcome the reviewers' insights and suggestions, which could guide the investigations further.

**C. Adaptive value of β.**

Dynamic adjustment of the β parameter, guided by an appropriate heuristic, has the potential to outperform any fixed value. In the rebuttal, we investigate this approach by allowing the teacher to adjust for a fixed number of steps initially (for a single loop), inspired by a recent influential Continual Learning paper [R1]. The results, presented in Table A.5, show that this approach yields slightly better outcomes compared to keeping the teacher fixed from the start. While this is not universally true across all settings, the findings highlight the promise of this method. Developing a more robust approach is left for future work, as determining the optimal timing for freezing the teacher requires careful tuning and hyper-parameter selection remains a challenging aspect of test-time adaptation.

[R1] Panos, Aristeidis, et al. "First session adaptation: A strong replay-free baseline for class-incremental learning." ICCV, 2023.

**D. The IT is only applicable to teacher-student framework-based methods.**

The EMA student-teacher models are a common approach in TTA and we show how it can have unwanted behavior in a reasonable extension of the current settings. Moreover, a universal SOTA method is not a contribution of this paper (See section A).

---

### Author Response · Authors · 2024-12-03

As the rebuttal period is nearing its end, we would like to summarize the current state of the rebuttal. Overall, the reviewers have acknowledged several strengths of the paper:

- **Extensive large-scale evaluations**: Conducted across diverse datasets, architectures and scenarios (iaLD, 3dWc) and loss functions (aKAL).
- **Problem importance**: Addressing the challenge of adaptation over long sequences (highlighted by reviewers aKAL and 3dWc) in practical scenarios (iaLD).
- **Broad applicability**: reviewers iaLD and 3kuq emphasized the versatility of our technique, which is applicable across various architectures and test-time adaptation methods based on the mean-teacher framework.
- **Simplicity and effectiveness**: as acknowledged by reviewers 3dWc and iaLD, with the latter also praising the method's robustness to changes in hyperparameters.
- **Presentation quality**: The paper received high scores for presentation, with the exception of reviewer 3kuq, who did not respond to our query regarding the reasons for their lower score.

The main weaknesses identified by the reviewers include the lack of theoretical justification (3kuq, iaLD), limited novelty (3dWc, aKAL), insufficient adaptive performance (3dWc, 3kuq), and concerns regarding hyperparameter selection (aKAL). During the rebuttal, we aimed to address the majority of the weaknesses raised by the reviewers through the following actions:

- **Successful application of IT**: We demonstrated its effectiveness on novel architectures (PETAL, as shown in Tables 3 and A.8).
- **Extensive hyperparameter search**: We explored additional tuning of hyperparameters (including CoTTA reset mechanism) (Tables A.3, A.4, and A.6).
- **Increased plasticity results**: We presented results where the teacher model was allowed to adapt for an initial fixed number of steps (Table A.5).
- **Novelty clarification with regard to RDumb paper**: We addressed this through detailed discussions with reviewer 3dWc and additional analysis in appendix (Section A.12).
- **Source domain data streams**: We provided results on adaptation to data streams without distribution shifts (Table A.7).

Finally, through further discussions with reviewers, we clarified:
- The necessity of using a fixed teacher model (instead of using student network only), addressing concerns from reviewer 3kuq.
- The use of batch statistics adaptation, as requested by reviewer 3dWc.

We would like to emphasize once again that the primary goal of the paper is the observation and analysis, rather than the IT method itself, which serves as a simple baseline for future work to "beat as a minimum." While we acknowledge the reviewers' point that the paper does not provide a theoretical explanation and ocassional underperformance on specific benchmarks, so do as many excellent machine learning papers that are empirically focused. We hope what we presented will spark discussion and inspire further investigation, maybe with more theorethical grounding as well. Most importantly, we believe the insights and performance analysis presented here will be valuable to the community and encourage further research on the limits of TTA methods.

Finally, we sincerely thank the reviewers for their initial feedback, which has significantly enhanced the quality of our work. We regret, however, that we have not received further responses from reviewers aKAL, 3kuq, and 3dWc following our latest revisions. This limits the opportunity for a more comprehensive discussion. However, we remain confident that the improvements implemented effectively address the concerns raised, and lead to higher scores from the reviewers whose feedback we have fully or partially addressed.

---

### Meta-Review · Area_Chair_saNn · 2024-12-22

**Metareview:**

The paper presents a method for test-time adaptation (TTA) based on the student-teacher framework proposed in some other recent works on TTA. The paper argues that, unlike these recent works, not changing the teacher weights performs better.

This paper received mixed score. The authors' response was discussed; however several of the reviewers' concerns remained. In particular, the insights reported by the paper are not very surprising (well-known in prior work) and the proposed solution doesn't seem to that competitive as compared to existing SOTA methods, raining concerns about the practical usefulness of the method. Therefore, the paper is slim in terms of providing new insights as well as providing a new method with improved performance.

In view of these concerns, the paper doesn't appear to be strong enough to be accepted.

**Additional Comments On Reviewer Discussion:**

The reviewer discussion raised several key points, such as insufficient experiments to validate the method (critical since the method itself is rather simple). The reviewers also remarked that similar observations have been made in prior works such as RDumb, the scope being limited to only TTA methods that are based on the student-teacher framework. Reviewers also felt that the paper should have expanded more on the investigation of Intransigent Teachers. These points are indeed relevant and lack of sufficient investigation weakens the paper's contributions further.

---

### Decision · Program_Chairs · 2025-01-22

Reject